# Investigating Dyslexia through Diffusion Tensor Imaging across Ages: A Systematic Review

**DOI:** 10.3390/brainsci14040349

**Published:** 2024-03-31

**Authors:** Bruce Martins, Mariana Yumi Baba, Elisa Monteiro Dimateo, Leticia Fruchi Costa, Aila Silveira Camara, Katerina Lukasova, Mariana Penteado Nucci

**Affiliations:** 1Laboratório de Investigação Médica em Neurorradiologia—LIM44—Hospital das Clínicas da Faculdade Medicina, Universidade de São Paulo, São Paulo 05403-000, Brazil; bruce.martins@hc.fm.usp.br (B.M.); mariana.yumi@hc.fm.usp.br (M.Y.B.); elisa.dimateo@hc.fm.usp.br (E.M.D.); 2Centro de Matemática, Computação e Cognição (CMCC), Universidade Federal do ABC, Santo André 09210-580, Brazil; leticia.fruchi@aluno.ufabc.edu.br (L.F.C.); camara.aila@aluno.ufabc.edu.br (A.S.C.); katerina.lukasova@ufabc.edu.br (K.L.)

**Keywords:** dyslexia, diffusion tensor imaging, structural brain changes, structural connectivity

## Abstract

Dyslexia is a neurodevelopmental disorder that presents a deficit in accuracy and/or fluency while reading or spelling that is not expected given the level of cognitive functioning. Research indicates brain structural changes mainly in the left hemisphere, comprising arcuate fasciculus (AF) and corona radiata (CR). The purpose of this systematic review is to better understand the possible methods for analyzing Diffusion Tensor Imaging (DTI) data while accounting for the characteristics of dyslexia in the last decade of the literature. Among 124 articles screened from PubMed and Scopus, 49 met inclusion criteria, focusing on dyslexia without neurological or psychiatric comorbidities. Article selection involved paired evaluation, with a third reviewer resolving discrepancies. The selected articles were analyzed using two topics: (1) a demographic and cognitive assessment of the sample and (2) DTI acquisition and analysis. Predominantly, studies centered on English-speaking children with reading difficulties, with preserved non-verbal intelligence, attention, and memory, and deficits in reading tests, rapid automatic naming, and phonological awareness. Structural differences were found mainly in the left AF in all ages and in the bilateral superior longitudinal fasciculus for readers-children and adults. A better understanding of structural brain changes of dyslexia and neuroadaptations can be a guide for future interventions.

## 1. Introduction

Dyslexia is a neurodevelopmental disorder that impairs fluency and/or speed of reading, as well as word recognition; it may also impact spelling. The difficulties should not be explained by other cognitive, health, or socio-economic factors [1]. The lack of accurate and fluent reading is a product of poor recognition and decoding abilities, and despite that, verbal comprehension is not equally affected in dyslexia [2,3].

Developmental dyslexia is classified within the framework of the diagnosis-specific learning disorder of reading [4], and it manifests in a spectrum from mild, to moderate, to severe [5]. It is present in over 80% of people with a learning disability in studied languages, having some variations regarding severity and type of difficulties according to the language structural characteristics [6]. Dyslexia’s mean prevalence is estimated at around 7% of the world population, with a predominance in male individuals [2,3]. Still, the incidence and prevalence are unclear due to the heterogeneity of literacy and language cultures with wide variances in terms of definitions, diagnostic instruments, rules, guidelines, and protocols for assessing dyslexic children and adults [1,7].

Two main medical classifications (ICD-11 and DSM-5) define the diagnostic criteria for dyslexia, and the assessment focuses mainly on identifying the reading and spelling discrepancies compared to the general population performance together with the assessment of other aspects that could confirm or rule out the diagnosis, for example, intelligence, phonological awareness, word and pseudoword reading, verbal fluency, verbal working memory, reading, and naming under stress [8]. This careful and time-consuming evaluation aims to support the clinical exclusion criteria, which contributes to reducing the prevalence by avoiding wrongful diagnoses of individuals struggling to read. Furthermore, there has been an effortful search in different fields for a better understanding of dyslexia’s development and neuroimaging contribution in clarifying neuroanatomical forming behind reading and dyslexia.

Clinical neuroimaging has been transformed by Diffusion-Weighted Imaging (DWI) and Diffusion Tensor Imaging (DTI), making it possible to examine the brain’s architecture and identify pathology earlier and more accurately than traditional magnetic resonance imaging sequences. Nowadays, diffusion is already used in clinical practice for stroke, trauma, tumors, demyelinating conditions, and neurosurgical planning. In psychiatric and neurological conditions, it is still used mainly in the research context [9].

The foundation of diffusion imaging lies in the behavior of water molecules, which move freely through space equally in all directions when unimpeded by structures. However, when encountering obstacles like cell membranes, water molecules tend to diffuse in alignment with the orientation of those barriers [10]. Magnetic resonance imaging, facilitated by DWI sequences, enables the measurement of water displacement in various directions for a brief duration. This information can then be used to assess tissue integrity, particularly in white matter fiber pathways [9].

Recent advancements in DTI research have expanded the focus beyond the assessment of a straightforward diffusion scalar to emphasize the significance of the more intricate 3D diffusion pattern. Axial diffusivity (AD), radial diffusivity (RD), mean diffusivity (MD), apparent diffusion coefficient (ADC), and others are additional imaging metrics that are increasingly used [11].

A standard DTI metric utilized in evaluating a variety of neuropathologic processes, from traumatic brain injury to demyelinating illness, is fractional anisotropy (FA), which quantifies the degree of this directionality [11]. Despite its promise for identifying subtle illnesses and alterations not discernible with conventional MRI sequences, the clinical applications of DTI have been questioned regarding the specificity of the findings [11].

In developmental neuroscience, DTI metrics have been shown to be a useful biomarker of white matter tract development and tissue injury, being used for treatment monitoring and potentially acting as outcome predictors. Children’s normal and pathological brain maturation has been described by the ADC/FA scalars since it is well known that as brain myelination and maturation advance, ADC values fall and FA values rise [12].

The literature has shown that people with dyslexia or reading disability may show brain structural changes with lower FA values in the left frontal and temporoparietal regions that coincide with the majority of studies on the left arcuate fasciculus (AF) and corona radiata (CR). Few studies have suggested a role for the posterior part of the corpus callosum or more ventral tracts like the inferior longitudinal fasciculus (ILF) or the inferior fronto-occipital fasciculus (IFOF) [13]. And more recently, a meta-analysis found no differences between dyslexics and typical readers when observing studies that conducted Voxel-Based Analysis (VBA) of FA and compared it to reading ability [14].

The purpose of this systematic review is to search for a better understanding of the multitude of possible methods for analyzing DTI data while accounting for the characteristics of a clinical population such as developmental dyslexia in the literature in the last 10 years.

## 2. Materials and Methods

### 2.1. Search Strategy

The systematic review searched the primary databases, PubMed and Scopus, for publications published within the last ten years, including the period from January 2011 to September 2022. The indexed articles were selected, and their findings were reported, following the Preferred Reporting Items for Systematic Reviews and Meta-Analyses (PRISMA) guidelines [15], and this study was not registered on Prospero. The criteria of interest selected were keywords in the following sequence: ((Dyslexia) AND (Brain connectivity) OR (Diffusion tensor imaging), using the boolean operators (DecS/MeSH):

SCOPUS: ((TITLE-ABS-KEY (dyslexia) OR TITLE-ABS-KEY (“Reading disorder”) OR TITLE-ABS-KEY (“Reading disorders”) OR TITLE-ABS-KEY (“Reading disability”) OR TITLE-ABS-KEY (“Reading disabilities” [) OR TITLE-ABS-KEY (“Developmental reading disability”) OR TITLE-ABS-KEY (“Developmental reading disabilities”) OR TITLE-ABS-KEY (“Developmental reading disorder”) OR TITLE-ABS-KEY (“Developmental reading disorders”))) AND ((TITLE-ABS-KEY (dti) OR TITLE-ABS-KEY (“Diffusion Tensor MRI”) OR TITLE-ABS-KEY (“Diffusion Tensor Magnetic Resonance Imaging”) OR TITLE-ABS-KEY (tractography) OR TITLE-ABS-KEY (“Diffusion Tractography”) OR TITLE-ABS-KEY (“Diffusion Tensor Imaging”) OR TITLE-ABS-KEY (“Diffusion weight imaging”) OR TITLE-ABS-KEY (dwi))) AND (LIMIT-TO (DOCTYPE, “ar”)) AND (LIMIT-TO (LANGUAGE, “English”)) AND (LIMIT-TO (PUBYEAR, 2022) OR LIMIT-TO (PUBYEAR, 2021) OR LIMIT-TO (PUBYEAR, 2020) OR LIMIT-TO (PUBYEAR, 2019) OR LIMIT-TO (PUBYEAR, 2018) OR LIMIT-TO (PUBYEAR, 2017) OR LIMIT-TO (PUBYEAR, 2016) OR LIMIT-TO (PUBYEAR, 2015) OR LIMIT-TO (PUBYEAR, 2014) OR LIMIT-TO (PUBYEAR, 2013) OR LIMIT-TO (PUBYEAR, 2012) OR LIMIT-TO (PUBYEAR, 2011)).

PubMed: (((((((((Dyslexia*[Title/Abstract]) OR “Reading disorder”[Title/Abstract]) OR “Reading disorders”[Title/Abstract]) OR “Reading disability”[Title/Abstract]) OR “Reading disabilities”[Title/Abstract]) OR “Developmental reading disability”[Title/Abstract]) OR “Developmental reading disabilities”[Title/Abstract]) OR “Developmental reading disorder”[Title/Abstract]) OR “Developmental reading disorders”[Title/Abstract]) AND ((((((DTI[Title/Abstract]) OR “Diffusion Tensor MRI”[Title/Abstract]) OR “Diffusion Tensor Magnetic Resonance Imaging”[Title/Abstract]) OR “Diffusion Tractography”[Title/Abstract]) OR “Diffusion Tensor Imaging”[Title/Abstract]) OR Tractography[Title/Abstract])).

### 2.2. Inclusion Criteria

The review included only original articles written in English published within the last 10 years, and full text available about dyslexia in structural analyses by DTI. According to the patient/problem, intervention, comparison, and outcome (PICO) criterion, the problem was: the structural brain changes in dyslexia are unclear; the intervention was: structural analysis by DTI; the comparison was: differences between dyslexia and typical readers volunteers; and the outcome was: the structural brain pattern of dyslexia of development.

### 2.3. Exclusion Criteria

We excluded studies based on the following criteria: (i) reviews or meta-analyses; (ii) publications written in languages other than English; (iii) indexed articles published in more than one database (duplicates); (iv) articles that included dyslexia with other neurological or psychiatric comorbidities such as stroke, brain injury, epilepsy, autism, traumatic brain injury, aphasia, and mood disorders; (v) articles that performed any analysis other than DTI, such as machine learning, graph theory, and only methodological comparison; (vi) articles in which the diagnosis of dyslexia is unclear; (vii) articles without at least one outcome or method of analysis reporting DTI measures and/or correlation with demographic or neuropsychological measures; (viii) case reports; (ix) neonates; and (x) dyslexia with genetic alterations.

### 2.4. Data Compilation

In this review, seven of the authors (B.M., M.Y.B., E.M.D., L.F.C., A.S.C., K.L., and M.P.N.), in pairs, independently and randomly analyzed, reviewed, and assessed the eligibility of titles and abstracts according to the strategy of established search. The authors B.M., M.Y.B., E.M.D., L.F.C., A.S.C., K.L., and M.P.N. selected the final articles by evaluating the texts that met the selection criteria. The authors B.M., E.M.D., L.F.C., and A.S.C. were responsible for the search for the demographic, clinical, and neuropsychological characteristics of volunteers and dyslexia patients, being checked by the senior authors (K.L. and M.P.N.). The authors B.M., M.Y.B., L.F.C., and A.S.C. searched for the characteristics of structural brain analyses and their outcomes, and all data were checked by the senior authors (K.L. and M.P.N.). All of the authors contributed to writing the entire text of this review.

### 2.5. Data Extraction

The selected articles were analyzed using two topics, which were represented in tables that addressed the following characteristics: (1) the demographic characteristics of the population sample, their language, their nationality, and the neuropsychological tools used to characterize the reading disorder; and (2) the characteristics of DTI acquisition, the parameters used in the image analyses and corrections, the structural outcomes between groups, and the correlation with clinical data when reported.

### 2.6. Risk of Bias Assessment

The selection of articles was performed in pairs, and a third independent author decided if the articles should be included. The data selected in the tables were divided by the authors into the groups already described above, and the checking of the data was carried out by the following group. The final inclusion of studies into the systematic review was by agreement of all reviewers.

### 2.7. Data Analysis

The data from the articles included in the tables were analyzed descriptively using the percentage, mean, and standard deviation; the variation to characterize each factor attributed to the demographic and neuropsychological characteristics of the participants in each study; and the characteristics of the acquisition, analysis, and results of the structural assessment performed by DTI image acquisition.

## 3. Results

### 3.1. Overview of the Screening Process of the Included Studies

Following the inclusion and exclusion criteria described above, we found 124 articles in the last ten years throughout the Scopus and PubMed databases, with 116 from Scopus and 8 from PubMed. Of the 116 articles found in Scopus, 64 were excluded after screening, two studies were with participants with alexia, one with aphasia, two with autism, one with mood disorder, two neonates, 20 without dyslexia diagnosis, two with genetic alterations, nine with brain injury (such as stroke, epilepsy, cortical lesion, and TBI), six were case reports, three were meta-analyses, seven were reviews, three were methodological studies, and six only featured morphometric analyses. The eligibility analysis excluded a further four articles. Three studies reported different methodologies of DTI analysis (machine learning, graph theory, and manual and automatic segmentation), and one did not report the DTI results, resulting in 48 studies included in this selection. Out of the eight articles identified in PubMed, five were excluded during screening: two lacked a dyslexia diagnosis, and three were duplicates from Scopus. After the eligibility assessment, an additional two studies were excluded. One study conducted DTI analysis using machine learning, while another did not report DTI results. Consequently, only one study from PubMed was included [16,17,18,19,20,21,22,23,24,25,26,27,28,29,30,31,32,33,34,35,36,37,38,39,40,41,42,43,44,45,46,47,48,49,50,51,52,53,54,55,56,57,58,59] in this systematic review, 48 were included from Scopus, and one was included from PubMed, as shown in Figure 1.

### 3.2. Demographic and Neuropsychological Characteristics of Studied Dyslexia Subjects

This systematic review aimed to provide an overview of dyslexia in various stages of life, from early childhood (preschool to kindergarten) with a familiar history of dyslexia to elementary school children, adolescents, and adults, following the structural brain changes involved in this neurobiological disorder. In the selected studies about dyslexia and structural brain analysis by tractography included in this systematic review, 15% of studies [16,17,18,19,20,21,22,23] included only children below 6 years old with the family risk of dyslexia before reading acquisition (classified as pre-readers); followed by 70% of the studies that analyzed older groups, at different stages of reading proficiency (classified as reading children) from 7 years old up to 19, males and females (classified as readers); and 15% of studied male and female adults aged from 20 to 33 years old (classified as reading adults) (Table 1).

Another important aspect is that dyslexia can manifest similarly across languages, but the characteristics and challenges may vary based on the language’s structure, writing system, and phonological rules. Because of this, we include the demographic characteristics of the country and the language participants spoke. The pre-reader studies were done mainly in the United States with English language speakers (50%) [16,17,18,22], followed by 40% [20,21,23] in Belgium with Dutch speakers, and 10% [19] in Germany with German speakers, including young children under 6 years old male and females and balanced distribution (a total of 193 females to 215 males) and with a sample variation of 10 to 46 children in each comparison group. In studies with reading-stage children, 47% spoke English—from the USA (45%) [25,27,31,32,33,39,42,44,48,50,52,53,54,56] and Canada (2%) [28,29]; 17% spoke French (the study was carried out in France [26,30,34,38,45]); 11% spoke Dutch—the studies were carried out in Belgium and Netherlands [37,41]; 8% were German; 6% spoke Mandarin—the studies were carried out in China [43] and Taiwan [36]; and 3% of studies were done with speakers of Arabic (Egypt) [35], Spanish (from Spain) [47], Portuguese (from Brazil) [49], or Italian (from Italy) [51]. In adults, the language of studies was less varied: 38% spoke English, and the studies were carried out in New Zealand [59] and the USA [62,64]; 25% spoke German (Germany) [58,60], 25% Dutch (Belgium) [61,63], and 12% Finnish (Finland) [57] (Table 1).

The neuropsychological characterization of the studied subjects followed the specificity of dyslexia diagnostic criteria, that the neuropsychological tests that were used included assessment of intelligence quotient (IQ); word and non-word reading and spelling tests; reading comprehension tests; rapid automatized naming (RAN); phonological awareness; and language, attention, and executive functions (description in Table 1). The primary purpose of the neuropsychological evaluation was to compare the performance of the dyslexic groups with the control group.

In the pre-reader, the intelligence was evaluated only by non-verbal tests, and only one study [16] showed a significant difference, with a worse performance of a risk of dyslexic children compared to the typically developing ones. One study did not report an intelligence assessment, possibly due to the very early age of participants [18]. In the reader children group, almost all studies evaluated the IQ (94.6%) with 51.4% by non-verbal and verbal IQ tests [25,26,27,30,31,38,39,40,45,47,48,49,52,53,54], 40.5% with only non-verbal tests [25,28,33,34,36,41,44,46,55,56], and 2.7% with only the verbal tests [50]. The significant difference between groups occurred in 34.3% of studies in verbal IQ tests and only 5.9% in non-verbal IQ tests. In the adult readers, 57.1% of the studies were administered non-verbal IQ tests [57,60,61,63,64] and only 28.6% showed significant differences with worse outcomes for dyslexic adults.

Concerning reading skills, in pre-readers, 89% [16,17,19,20,21,22,23] of studies assessed the children by word, letters, or pseudoword reading, and out of these 44% [16,17,19,21,22,23] had significantly lower outcome compared to the control group. In this age range, the RAN was also generally employed (78%) [16,17,19,20,22,23], with objects and colors being the most commonly used items (56%). A significant difference was found in naming speeds, mainly for slower naming of objects by dyslexic children in 44% of studies of this population. The phonological awareness was also mostly assessed (75%) [16,17,19,20,21,22,23], and in 25% of studies performance was lower in dyslexic groups. As for the other cognitive functions, attention was assessed in just one study [19], and the findings on working memory, digit span, visual reception, and gross and fine motor were also seldom reported.

Regarding the reader children, in all of the studies the reading skills were assessed by the word reading, and as would be expected lower performance of children with dyslexia compared to controls was found in 79% of studies [24,27,30,31,32,33,34,38,39,40,41,42,43,44,45,46,47,48,49,50,51,52,53,54,55,56]. Other significant between-group differences with worse outcomes for dyslexic children were reported for pseudoword reading (70% [24,27,28,30,31,32,33,34,38,39,40,41,44,45,46,47,48,51,52,53,54,55,56] out of 85% assessed studies) and text reading (39% [24,30,32,34,39,40,45,46,51,52,54,55,56] out of 64% assessed studies). In 45% of all studies, the phonological awareness was assessed [24,26,27,31,34,35,38,39,40,41,43,45,46,51,53], and 39% of studies [24,26,27,31,34,38,39,40,41,43,45,46,51,53] produced significant results when compared between groups. When assessing RAN, 39% of studies evaluated this ability [24,26,29,34,35,38,40,41,43,45,46,52,56], and 33% found a significantly worse outcome for the dyslexic children when compared between groups [24,26,34,38,40,41,43,45,46,52,56]. A small number of studies (9%) assessed language [38,43,56] and attention [24,27,40]; however, almost all of these studies showed worse performance of subjects with dyslexia when compared to the controls. Half of the studies reported results of other cognitive functions in different aspects of short-term memory (digit span 33.3%) [24,26,27,38,40,45,46], verbal working memory (33.3%) [24,26,38,40,43,46], working memory (17%) [41,47], and arithmetic or mood behavior (11%) [46,56]. Almost all of these assessments also reported significant between-group differences and lower outcomes for participants with dyslexia.

For the reading adults, 100% of studies assessed volunteers for the word and/or pseudoword reading [57,58,59,60,61,62,63,64]. In 37.5% of the studies [57,58,62,64], text reading was tested, and all tests produced significant findings that allowed the groups to be distinguished based on the lower performance of the dyslexic group. Only 50% of these studies that tested RAN and phonological awareness showed significant group differences [57,59,63,64], and 25% assessed language and attention domains [57,59], without significant group differences.

Considering the neuropsychological outcome of reading children in the three language groups most represented in this revision (English, French, and Dutch), there was no difference in the cognitive skills found impaired in dyslexic volunteers compared to the controls.

### 3.3. Brain Structural Connectivity Characteristics on Acquisition, Process, and Outcomes of Dyslexia

The structural analysis through the DTI acquisition was performed in 88% of studies in high-field MRI equipment (3T) (scanner by manufacturers: Siemens (Berlin, Germany) (49%) [16,17,18,19,22,24,25,26,30,31,32,34,36,38,40,41,43,44,45,46,55,57,58,60], Philips (Eindhoven, Netherlands) (35%) [20,21,23,27,33,37,47,48,50,51,52,53,54,61,63,64], General Electric (GE, New York, United States of America) (4%)) [28,56] and 10% in low-field (1.5T) (scanner by manufacturers: Siemens (6%) [39,59,62], Philips or GE (2%) [35,49]) used in the acquisition of older participants such as reader children (more than 7 year old) and adults (more than 18 year old), as shown in Table 2.

Most of the selected studies (91.8%) used the sequence DTI for the diffusion analysis, and only 6.1% used diffusion kurtosis imaging (DKI) protocols [25,32,33] that require at least 3 b-values (as compared to 2 b-values for DTI) and at least 30 independent diffusion gradient directions (as compared to 6 for DTI), in which these 3 b-values were used: 0, 700 or 800 or 1000, and 2000 s/mm^2^ with 30 or 32, and 64 noncollinear diffusion directions. In two studies, the DTI sequence also was reported with 3 b-values (0, 700, or 1000, and 1000 or 2000 for b-values with 64 diffusion directions), 50% of DTI studies used 2 b-values (0 and 700 or 800 or 1000 or 1400) and 37% studies only one b-value (700 or 800 or 1000 or 1300 or 1400 or 5000). Regarding the number of noncollinear diffusion gradient directions in the studies that acquired DTI sequence, 41.3% reported more than 60 directions (28.3% was 60 and 2.2% was 128 directions), 34.8% used between 30 to 56 directions, 13% of studies used less than 30 directions (the smaller number of directions was 6), and 10.9% did not report this parameter [46] (Table 2).

The basic pulse sequence repetition time (TR) and echo time (TE) parameters ranged from 3000 to 14,000 ms and from 55 to 110 ms, respectively; the slice image ranged from 23 slices with 5 mm of thickness to 160 slices with around 1.7 mm to cover the entire brain, and the field of view (FOV) ranged from 208 to 282 mm.

The DTI analysis was performed in the selected studies by different softwares, and usually (82%) used more than one software to conduct all of the analysis. Most of the studies (63%) used the FSL software [18,19,23,24,25,26,28,30,31,32,33,38,39,40,42,43,44,45,46,47,48,49,50,52,53,55,58,59,60,61,63] and its different tools (Neurite orientation dispersion and the density imaging (NODDI) model, Tract-Based Spatial Statistics (TBSS), FDT, DTIFIT, PR0OBTRACKX, and BEDPOSTX, among others) associated or not with other software; 18% of studies [16,17,18,21,22,27,32,40,41] used the VistaLab developed at Stanford University that comprises different tools such as MrDiffusion and Automated Fiber Quantification (AFQ); 20% used ExploreDTI [20,23,24,29,37,38,43,45,61,63]; 16% used the TrackVis [18,20,23,24,26,37,43,45]; 12% DTIprep [16,17,18,22,44,50], 10% MRTrix [19,25,32,40,57]; 10% SPM [41,51,56,62,64]; 6% TRActs Constrained by UnderLying Anatomy (TRACULA) [33,44,52]; 4% DSI Studio [36,57]; and 2% BrainVoyager [34,51], the only commercial software; and few studies used PANDA, DIPY, Reproducible Objective Quantification Scheme (ROQS), TORTOISE, and TractSeg.

Artifacts in DWI acquisitions lead to errors in tensor estimation, and Eddy Current (EC) distortions and Head Motion (HM) are the two primary intrinsic DTI acquisition abnormalities that may obliterate the voxel-wise correlation across all the DWIs. Most of the selected studies (73%) reported in the pre-processing step comprise the EC and HM (with a cutoff from 1.5 mm to 6 mm) corrections. Other studies (10%) reported corrections to the image (EPI) distortions, 4% for the Gibbs artifact (truncation or ringing artifact) or Marchenko–Pastur Principal Component Analysis (MP-PCA), and only 14% did not report any correction in the preprocessing image step.

Some FSL tools were used for these corrections such as CATNAP (Coregistration, Adjustment, and Tensor-solving a Nicely Automated Program), which is a data processing pipeline for Philips PAR/REC Magnetic Resonance data files, performing motion correction for both diffusion and structural images using FSL FLIRT; it adjusts the diffusion gradient directions for scanner settings (i.e., slice angulation, slice orientation, etc.) and motion correction (i.e., the rotational component of the applied transformation) and computes tensor and derived quantities (FA, MD, colormaps, eigenvalues, etc). Also, the TOPUP tool of FSL is used to correct images of the susceptibility-induced distortions (fix EPI distortions), and QUAD (Quality Assessment for DMRI) for automatically performing image quality control (QC) at the single subject.

The tracking of DTI group analysis was normally reported as whole brain, tract, or ROI-based, from voxel-based, and 34.7% of studies reported as ROI-based methods [17,18,20,21,22,34,35,37,39,42,48,50,52,53,54,60,63], 26.5% as whole brain and ROI-based methods [16,19,23,27,28,29,30,31,33,41,43,45,47], 22.4% only whole brain or fiber tract-based analysis [24,25,26,32,36,38,40,44,46,57,64], and 12.2% as voxel-based analysis [49,51,55,58,59,62].

Regarding DTI quantitative analysis, 92% measured FA; of these, 29% also reported AD or RD [18,22,31,39,42,43,44,45,46,47,48,49,50,54,61], and 22% MD. In a few studies, other anisotropy metrics were used such as 4% relative anisotropy (RA) [42,50], 2% QA [57], or hindrance-modulated orientation anisotropy (HMOA) [45]. The MD measure was also described by directionally averaged mean diffusivity (Dav) [54,64] and extra-axonal mean diffusivity (MDe) [32], in 4% and 2% of studies, respectively. The ADC [35] or exponential apparent diffusion coefficient (eADC) metrics were reported in 2% of studies. Some White Matter Tract Integrity (WMTI) metrics from DKI were reported in 2% of studies, such as axonal water fraction (AWF) [32], intra-axonal diffusivity (Da) [32], or MDe [32], and NODDI metrics such as Neurite density index (NDI) and Orientation dispersion index (ODI) were reported in 4% of studies each [31,33].

Different types of atlas were reported in the selected studies to segment the cortical and white matter region; 54% used one of FSL atlas such as Julich-Brain Cytoarchitectonic Atlas, MNI (Montreal Neurological Institute) Structural Atlas, JHU (Johns Hopkins University) ICBM-DTI-81 White-Matter Tractography Atlas, and Harvard-Oxford atlas; 19% used one of three FS’s atlas (Desikan–Killiany, Destrieux, and Desikan–Killiany–Tourville cortical atlas); 9% used native space; 7% used the automated anatomical atlas (AAL), the template for SPM, AFQ, ExploreDTI, and PANDA software; 3% used Talairach atlas; and 9% did not report this information.

Most of the studies (90%) specified the tracts/ROI used to explore the group comparison or association between structural data and demographic or neuropsychological data, the main tracts were AF (49%) [16,17,18,19,20,21,22,23,25,27,32,34,35,37,40,41,43,44,45,47,51,59,61,63], inferior longitudinal fasciculus (ILF) (41%) [17,22,25,27,32,34,37,39,40,41,42,43,44,46,48,49,50,51,53,62], inferior frontal-occipital fasciculus (IFOF) (39%) [20,23,25,32,34,36,37,39,40,41,43,45,47,49,50,51,62,63,64], superior longitudinal fasciculus (SLF) (37%) [16,17,22,25,27,31,32,35,39,40,41,42,44,45,47,50,51,56], 24% for corpus callosum (CC) [17,25,36,39,47,51,52,53,54,55,61,62], 18% for corticospinal (CS) tract [18,32,35,39,40,41,42,49,50], 12% for CR (including anterior and posterior parts—aCR and pCR) [35,39,55,59,62,64], 20% for uncinate fasciculus (UF) [24,25,32,37,39,40,41,43,49,50], and 16% for thalamic radiation (ThR) (including anterior and posterior parts- aThR and pThR) [32,36,40,41,49,50,52,62]. Forceps minor (FMn) and major were reported in 10% of studies [32,40,41,49,50], as well as the cingulum (CG) [31,40,42,49,50]; less frequently was also reported in 4% optical radiation (OR) [42,51], as well as cerebellar peduncles [36], internal capsule [62], temporal and temporo-occipital regions [19], frontal aslant tract (FAT) [24], primary auditory cortex, lateral geniculate nucleus (LGN) [60], and inferior colliculus (2% each). 6% of studies reported only the atlas used [26,29,30], did not specify the tracts or ROIs, and 4% did not report this information [46,57].

Regarding the results of the structural analysis of the brain, 89% of the studies described this finding, with the predominance of a decrease in FA in the left hemisphere, when comparing the dyslexia group with the control group, and 10% of the studies did not show a significant difference between the groups [19,25,46,59,64], occurring mainly in the group of adults (25%) [59,64].

The tractography analyses of the pre-reader group reported by the selected studies were based on the FA changes according to the risk of familial history of dyslexia (FHD); 13% reported higher FA of right SLF [16] or CC [17] compared to the children with positive FHD than TR negative FHD, and 50% showed lower FA of the left AF (as also long portion) [18,21,22,23] and 13% in IFOF [20], as shown in Figure 2, highlighting the frequency in green color. Almost all of these regions also showed a positive correlation with age and some language tests and predicted decoding skills or reading impairment.

In the reading children, 42% of these studies reported a significant change in FA values (27% lower [27,38,42,43,44,45,47,49,55] and 15% higher values [40,41,48,52,54] in dyslexic children than the TR group), decreasing mainly in the left tracts such as AF, SLF, ILF, CR, and IFOF and increasing in the right SLF, aThR, cingulum, and CC. These studies also showed 6% of AD [43,50] decreased in dyslexic children in IFOF, SLF, UF, FMn, CG, and the right ThR [50], as well as in the left AF and ILF [43], and 12% of MD [28,32,39] changes between groups (6% higher [28,32] and 6% lower values in dyslexic children [32,39]). Only 3% of these studies reported higher values of ODI in the left ventral optical tract [33] of DYX and lower values of RD and HMOA (in UF of DYX males) [24], as shown in Figure 2, highlighting the frequency in red color. No group differences were reported in 6% of these studies [25,46], and 30% did not report brain changes [26,29,30,31,34,35,36,37,51,56]

In the selected studies with adult participants, 50% showed group differences, in which the DYX had low FA in lateral geniculate nucleus, TOC, and left-AF [60,63], and another DTI metric was the QA with high values for DYX group in left- UF, CS, CC, ThR, FMj, and parietal tracts and low in left SLF, CS, and OF in comparison to TR group [57], as shown in Figure 2, highlighting the frequency in purple color. In total, 25% of these studies did not find a group difference [59,64] or report this information [61,62].

The outcomes were also reported by the correlation analysis between clinical, demographic, and neuropsychological data, with white matter changes in 83.6% of studies, in which 60.9% showed a positive correlation between FA/MD changes and age (12%) [18,25,39,41,48,54]; gender (4%) [25,29], mainly in children; and neuropsychological tests (phonological awareness, word/no-word identification skill, VAS, working memory, verbal fluency, digit span, and Chinese character recognition) (44.9%) [20,22,24,25,27,29,30,31,32,35,36,37,38,39,43,44,46,54,56,57,61,63] in all ages; 32.6% reported a negative correlation between FA/MD/RD changes and neuropsychological performance [28,30,32,34,35,37,39,40,43,44,45,46,52,57,60,61]; 5% of studies reported [19,21,23] changes between white matter and DYX diagnoses or neuropsychological outcomes/skills; and 16.4% did not find or report correlation results [33,42,47,49,50,53,55,59].

## 4. Discussion

This systematic review provided an overview of the main structural brain changes findings by the DTI technique in developmental dyslexia in different age groups, including early at-risk children due to a family history of the disorder, as well as children and adults with reading disorders who have been diagnosed with dyslexia. By comparing the results of several studies, we describe converging evidence on structural abnormalities of the brain and the associations between imaging results and changes in this population’s characteristics. This systematic review observed that the assessment of different cognitive functions by the neuropsychological instruments of the selected studies varied according to age, country, and language; however, the lack of standardized procedures among age and language groups drastically reduces the possibility of comparing the behavioral outcomes and its relation to the structural brain changes. While some aspects of reading, mainly word decoding and recognition, are the most assessed skills in all the age groups, others such as language, attention, and working memory are reported exceptionally. The intelligence measure showed an expected outcome in terms of experimental and control group matching, with more studies reporting lower performance of children with dyslexia on verbal but not on non-verbal tests of intelligence. The intelligence assessment has generated a long-time debate in the field of learning, and while in typically developing children intelligence performance generally correlates with the achievement level, in reading impairment the intellectual disability is inconsistent with a diagnosis of dyslexia [65].

The importance of matching groups on intelligence measures is also supported by a growing body of evidence linking the brain white matter organization to intelligence level and processing efficiency. A recent study showed stronger integration of white matter structures within a local community (ex. frontal region) and especially with external brain networks (ex. frontal to parietal regions) in adults scoring high on non-verbal tests compared to average performers [66].

The structural brain changes shown by the DTI measurements were examined or extracted using the atlases, and nearly all studies (92%) examined the fractional anisotropy, which represents the directionality and organization of tissue microstructure; other studies examined some of its variants, including RA, QA, and HMOA, providing additional information to FA. The studies also examined the following measures: the AD measure, which can show changes in the density or integrity of axons within white matter pathways; the RD measure, whose increase frequently denotes disruptions in the microstructure of white matter, such as demyelination or axonal damage; and the MD measure, which, along with some variations like Dav and MDe, typically shows higher values that indicate decreased tissue integrity and increased diffusion.

In children before reading acquisition who had a positive family history of dyslexia, the structural changes analyzed by DTI metrics were shown mainly by the FA alterations. The decrease in FA was predominant in the left of the AF, as well as for IFOF, a result also reported in a recent study with dyslexic children with this profile [21], and a pattern of increased FA occurred in few studies, and only in the right hemisphere of the SLF and sCC, which may suggest possible early neural compensatory mechanisms in the right hemisphere [17]. This pattern was also similar in reading children, with low FA values mainly in the left hemisphere involving the AF, but also high FA values in the right hemisphere of AF, showing more neuroplasticity signals than the other young group. In addition, in these reader groups the structural changes covered more areas, still with a predominance of the left hemisphere, and were identified in other DTI metrics, such as low AD values in AF, SLF, IFOF, UF, ThR, ILF, CG, and CS, as well as a high MD values in AF, and low in UF and CR. Children submitted to reading intervention show an increase in MD values in AF and other reading brain circuitry such as the left ILF and posterior CC [67]. Variations of anisotropy, such as RA and HMOA, also showed low values in brain structural changes reported in the articles with reader children, but only FA showed high values in the following tracts AF, SLF, CC, ThR, ILF, and CG.

In dyslexic adults, the percentage of structural changes reported was much lower than those reported in children. It seems that with brain development, in adulthood, the structural differences of dyslexia become less evident due to neuroadaptation. However, the pattern of decreased FA measurements in dyslexic adults remained the same, with predominance in the left hemisphere of AF and the region that comprises the AF (lateral geniculate nucleus, and temporo-occipital cortex), as well as bilaterally SLF, CC, CS, and the left of UF and ThR by the QA measurements. This anisotropy variation, the QA measure, also showed an increase in the left side of SLF, VOF, and CS, which may represent neuroplasticity in adulthood. Nonetheless, due to literature scarcity on dyslexia in adults it is difficult to compare these results found in the review with other studies.

Another form of result widely explored between studies was the analysis of association, be it through correlation or the prediction of structural data with demographic or neuropsychological data, and this occurred with a greater incidence of significant findings between studies than the actual comparison of structural changes. These association analyses are normally applied to assist in the interpretation of results, mainly in studies with adults when there were no structural difference between the groups, but measures mainly of FA were positively or negatively correlated with the results of neuropsychological tests. In children, the structural changes helped predict some performance in neuropsychological tests, especially in children with a family risk of dyslexia.

In addition to the structural results found in participants with dyslexia and controls, this review considered how DTI data were acquired, processed, and extracted as an analysis. In general, all studies included in this review took good care to ensure good image and data quality, reducing bias in the interpretation of results.

The DTI acquired in high magnetic field equipment, such as 3 Tesla, can increase the capacity to acquire higher resolution scans more quickly, with higher b-values and thinner slices, as well as to increase tissue contrast and reduce background noise (thereby increasing the signal-to-noise ratio and contrast-to-noise ratio) [68]; this magnetic field was used in 88% of the studies (49% Siemens, 35% Philips, and 4% GE), in the acquisition of both DKI and DTI sequences to study tractography. The latter sequence is the more traditional sequence and was used in most of the selected articles in our systematic review. The difference between the two is that DKI significantly reduces the error of dODF orientation estimates compared to DTI and makes it possible to detect crossing fibers, which leads to a noticeable improvement in tractography across regions with complex fiber bundle geometries [69,70]. DKI-based tractography has potential benefits, especially in clinical contexts when time is of the essence [71].

The acquisition parameters of the DTI is important because they affect values of white matter (WM) scalar metrics, including FA, MD, Signal Noise Ratio (SNR), and even entire brain tractography investigations. These acquisition parameters include the diffusion sensitivity coefficient (*b*-value), which is a factor that reflects the strength and timing of those gradients used to generate DWI, as well as the reliability of DTI results concerning image and data quality; diffusion directions, in which there are more directions the longer the acquisition time; and voxel size (the smaller the size, the higher the quality [72,73]).

The adequate b-value for diffusion imaging quality evaluation in dyslexia was unclear and varied according to the equipment’s magnetic field. In a heath brain, a greater b-value usually results in a poorer SNR and image quality because increased signal attenuation owing to diffusion as well as increased TE (and therefore additional signal loss due to T2 decay), would lead to the decrease of MD, AD, and RD when the gradient directions and voxel resolution remained constant [73,74]. In the literature, this is more evident at the low field strength of the MR scanner as 1.5 T [75], and only 10% of the selected studies of this review used this field, less evident in high field strength (3 T and 7 T) due to their relatively high SNR, in which the increase in b value has little influence on the decrease in SNR, showing the best image and data quality with the respective b-values, 200 and 900 s/mm^2^ [73]. The selected studies included in this review reported normally more than one b-value and the MRI of 1.5T used b-values ranging from 800 to 1000 s/mm^2^ and the 3 T from 700 to 5000 s/mm^2^ [73,74].

The increased number of diffusion-encoding gradient directions can also improve DTI quality by averaging and strengthening the tensor estimation, and the opposite can reduce all DTI scalar values’ accuracy and precision; a minimum of 18 diffusion directions is advised to produce trustworthy DTI scalar results using the TBSS toolbox of FSL software [76]. In our study, just a small number of the selected studies (11.8%) used fewer than 30 directions, and of these, 7.8% used less than 18 directions, whereas the majority (78.4%) used 30 directions or more (41.3% used more than 60 directions).

Considering the impact of MRI acquisition parameters on the values of DTI measures, changes in the number of gradients and voxel resolution have the greatest impact on the FA, but variations in the b-value have a special effect on MD [72]. Another study also showed that the number of gradient directions was more relevant than the spatial resolution in some quantitative measures of DTI, such as tract volume, median fiber density, and mean FA, but this did not occur for all tracts evaluated in the same way, only for SLF and IFOF [77].

One of the most defining and defiant components of building a tracking algorithm is determining the underlying model that connects the raw dMRI images to the local fiber orientations, and presently, there is a wide range of software packages that incorporate higher-order fiber-tracking techniques that can calculate the relative contributions and orientations of several fiber populations within each voxel, which are easily applied to clinically relevant data sets [78].

A wide range of processing functions are offered by these software packages, such as tensor calculations, fiber tracking, visualization, statistical analysis, quantitative measure extraction from DTI datasets, and integration with additional neuroimaging tools. They vary, though, when it comes to the tractography algorithms that are applied, such as probabilistic, which produces a vast collection or distribution of potential trajectories from each seed point, and deterministic, which assumes a unique fiber orientation estimate in each voxel [79]; as well as the local approach, which is a fast and widely used method, it follows the local orientations of previously extracted fibers independently of each other, but the sum of small errors in these local orientations can significantly affect the final result, making it a very weak predictor of data with little quantitative significance or biological [78]. On the other hand, the global methods offer improved stability concerning noise and imaging artifacts and a greater agreement with the real dMRI data that was recorded; however, they rely on stochastic optimization approaches and hence do not guarantee convergence to a globally optimal solution [80].

The choice of software to analyze the DTI normally depends on the specific analysis requirements, familiarity with the software, and preference for user interface and workflow, and the studies normally used more than one software to employ specific tools to conduct the entire analysis.

Several image adjustments are usually conducted before DTI analysis to enhance the data quality and reduce common artifacts that may occur during data acquisition, allowing for appropriate interpretation and trustworthy outcomes. Most of the adjustments reported by the studies were Eddy current correction (74%) which aligns the DWI to a reference image acquired without diffusion weighting [81], and motion correction (74%); few explore structural analyses but they realign the image to compensate for subject motion, ensuring that the diffusion measurements are accurate and consistent across the dataset [82]. However, in low frequency the Gibbs ringing correction (4%) of the discontinuities in the k-space data caused by undersampling during image acquisition was also reported, resulting in obscure anatomical structures and affecting diffusion measurements, as well as EPI distortion correction (10%), considering the spatial variations in the strength and direction of the magnetic field gradients used for diffusion encoding. Unhappily, 14% of the studies did not report any adjustment before DTI analyses.

Regarding the type of quantitative DTI analyses reported by the studies, most (69.2%) used voxel-wise, which analyzes each voxel to identify differences in diffusion properties between groups or conditions, providing more detailed spatial information about diffusion metrics at a voxel-level; this is present in some of the software such as FSL, SPM, MRItrix, and ExploreDTI. Interestingly, a lot of articles reported this analysis as the whole brain since voxel-based analysis includes all the cerebro voxels. Also, ROI-based analysis used in 61.5% of studies is frequently conducted after voxel-based, making it challenging to determine which method was used in each study.

The outcome description of the studies was based on anatomical regions of interest or specific tracts outlined by a well-established atlas that facilitates the interpretation of imaging data by providing standardized anatomical labeling and spatial coordinates. While the atlases share the goal of delineating brain structures and regions, they differ in several aspects, including their origins, resolutions, and intended applications.

The study’s limitations stem mainly from the differences in how dyslexia was defined across studies. The use of the terms dyslexia, developmental dyslexia, reading disorders, or never reading difficulties point to the lack of a common terminology and diagnosis criteria. The impact of the findings of this study could be that the results may be possibly drawn from a very heterogeneous sample. However, the cognitive testing of participants may ameliorate this and emerge as a potential tool mainly in internationally normed tests.

Another limitation of the study was the influence of language on dyslexia’s reading acquisition history. It is a well-known fact that readers in irregular language systems have longer reading acquisition and struggling readers may develop different compensation strategies.

Studies on neuroimaging may be limited by variations in image acquisition parameterization; however, despite this wide range, all studies used optimal acquisition and analysis parameters that did not affect the comparability of results, even in cases where certain fundamental information was not stated. In terms of results, a certain study focused on describing the anatomical regions examined rather than the tract as most studies did, taking into account the tracts involved in the regions described.

## 5. Conclusions

This systematic review of structural alterations in the brain associated with dyslexia revealed that over the past ten years studies on children have outnumbered those on adults, primarily focusing on boys and the English language. The studies also showed that brain changes concentrated in FA reduction in the fasciculus arcuate of the left hemisphere at all ages, and in the left superior longitudinal fascicle for reading in children and adults, as well as an increase in the right hemisphere, which may indicate signs of neuroadaptation. A better understanding of structural brain changes of dyslexia and neuroadaptations can be a guide for future interventions.

## Figures and Tables

**Figure 1 brainsci-14-00349-f001:**
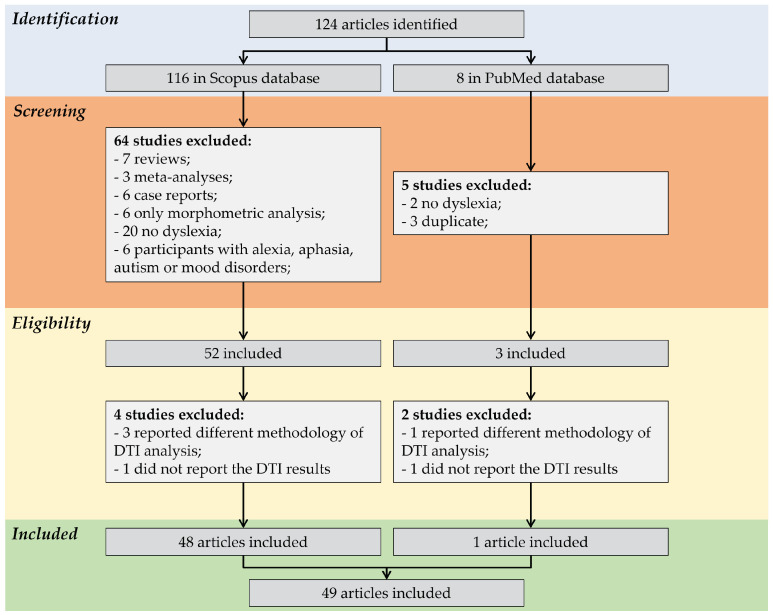
PRISMA flowchart of this systematic review study, identifying at each stage the number of studies included and the reasons for excluding studies until the final stage of inclusion of studies.

**Figure 2 brainsci-14-00349-f002:**
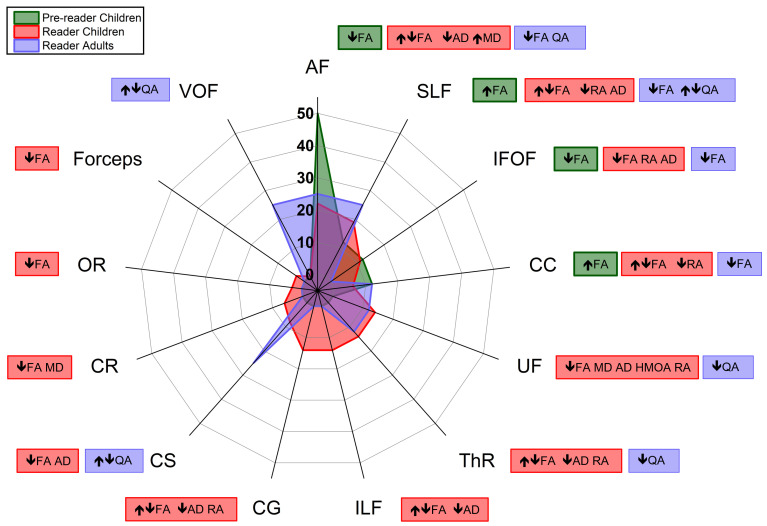
Spider graphic of the diffusion tensor image (DTI) outcomes percentage distributed by the main tracts reported in the systematic review and their DTI metrics behavior found according to tract and age group of dyslexia participants (pre-reader children in green, reader children in red, and reader adults in purple). The arrows indicate the increase or decrease in DTI metrics in the dyslexia group in comparison to the control group. Abbreviations: AF: arcuate fasciculus; SLF: superior longitudinal fasciculus; IFOF: inferior fronto-occipital fasciculus; CC: corpus callosum; UF: uncinate fasciculus; ThR: thalamic radiations; ILF: inferior longitudinal fasciculus; CG: cingulate cortex; CS: corticospinal fasciculus; CR: corona radiata; OR: optic radiation; Forceps: forceps major and minor; VOF: ventral occipital fasciculus; FA: fractional anisotropy; AD: axial diffusivity; MD: mean diffusivity; QA: quantitative anisotropy; RA: relative anisotropy; and HMOA: hindrance-modulated oriented anisotropy.

**Table 1 brainsci-14-00349-t001:** Demographic and neuropsychological assessment.

Ref.	Year	Country	Language	Group	N	Sex (F:M)	Age (Years)	Years of Education (or Level)	IQ	A Word Reading/Spelling	Pseudoword Reading	Text Reading	RAN	Phonological Awareness	Language	Attention	Others
Zuk J, et al. [16]	2021	USA	English	TR FHD− TR FHD+ RD FHD+	39 18 17	21:18 7:11 9:8	5.5 ± 0.3 5.5 ± 0.3 5.7 ± 0.4	Pre-Kindergarten; Kindergarten	**Non-verbal (KBIT-2)**	**LWID (WRMT-R/NU and WRMT-R); Letter Sound Knowledge (YARC)**	NR	NR	**Objects;** **Colors; Letter**	**Elision;** **Blending Words** **(CTOPP)**	Vocabulary Knowledge (PPVT-4); Sentence Comprehension (CELF-4); Speed Accuracy	NR	WM: Nonword Repetition (CTOPP), Sentence Repetition (GAPS)
Yu X, et al. [17]	2020	USA	English	TR FHD− TR FHD+ RD FHD+	34 35 12	16:18 17:18 4:8	5.4 ± 0.3 5.5 ± 0.4 5.8 ± 0.5	The end of 1st grade to 4th/ grade	Non-verbal (KBIT-2)	**WID (WRMT-R);** **SWE (TOWRE-2)**	**PDE (TOWRE-2); ** **WA (WRMT-R)**	NR	**Objects; Colors**	CTOPP	CELF-4	NR	HLE
Langer N, et al. [18]	2017	USA	English	FHD+ FHD−	14 18	7:7 10:8	0.9 ± 0.3 0.8 ± 0.3	NR	NR	NA	NA	NA	NA	NA	Expressive and receptive language (MSEL)	NA	Gross and fine motor (MSEL); Visual reception (MSEL)
Kraft I, et al. [19]	2016	Germany	German	FHD+ FHD−	25 28	11:14 12:16	5.7 ± 0.4 5.6 ± 0.4	Kindergarten	Non-verbal	**One minute** **word reading (SLRT-II);** **Spelling (DERET)**	**One minute** **pseudoword reading** ** (SLRT-II)**	**NR**	**Subtest (BISC)**	Pseudoword repetition (SETK 3-5), SS and RI (BISC), and PA (BAKO)	NR	Symbol comparison (BISC)	DS (K-ABC)
Vandermosten M, et al. [20]	2015	Belgium	Dutch	FRD+ FRD−	36 35	13:23 17:18	5.1 ± 0.2 5.1 ± 0.2	The last year of kindergarten	Non-verbal (Raven)	Letter knowledge productive/receptive test	NR	NR	Objects; Colors	End-phoneme and end-rhyme identification task (PA)	NR	NR	NR
Van Der Auwera S, et al. [21]	2021	Belgium	Dutch	*PreR: FRD−* *FRD+* *BR: FRD−* *FRD+* *FR: FRD−* *FRD+*	24 16 13 24 10 15	14:10 8:8 13:10 12:12 2:8 5:10	6 ± 0.1 6 ± 0.1 8 ± 0.1 8 ± 0.1 11 ± 0.1 11 ± 0.2	Kindergarten1st/ 2nd grades 3rd/4th/5th grades	Non-verbal (Raven and Block Design– WISC-III)	**Word Reading** **List; Letter** **knowledge; and** **Spelling**	**Pseudoword Reading Test**	NR	NR	**Phoneme Deletion and Spoonerism; PA**	NR	NR	NR
Wang Y, et al. [22]	2017	USA	English	*PreR: FHD−* *FHD+* *BR: FHD−* *FHD+* *FR: FHD−* *FHD+*	16 24 23 24 10 15	8:8 14:10 12:12 13:10 2:8 5:10	5.3 ± 0.8 5.4 ± 1.1 7.0 ± 2.2 7.3 ± 2.2 10.0 ± 2.5 10.2 ± 1.8	Nine single words 1st/2nd grades 3rd/4th/5th grades	Non-verbal (KBIT-2)	**WID (WRMT-R)** **(FR); SWE (TOWRE)** **(BR and FR); and** **TOSWRF (FR)**	**PDE** **(TOWRE)**	**Gray Oral Reading Test(GORT-5), Reading Fluency WJ-III-TA**	**Objects (PreR);** **Colors**	**CTOPP (FR);** **PC (WRMT-R, BR)**	**CELF-4 (BR)**	NR	**TOMAL2 (FR)**
Vanderauwera J, et al. [23]	2017	Belgium	Dutch	DYX TR FRD+ FRD−	15 46 34 27	7:8 15:31 13:21 9:18	7.9 ± 0.1 7.9 ± 0.1 7.9 ± 0.1 7.9 ± 0.1	PreR—prior 1st grade/ BR—2nd/3rd grade	Non-verbal (WISC)	**Word reading, one-minute test (BR); Spelling (BR); and Productive/Receptive Letter Knowledge (PreR)**	**Pseudoword reading** **two minute** **test (BR)**	NR	**Objects (PreR);** **Colors**	**PA (PreR);** **End-phoneme and** ** end-rhyme** **identification task**	NA	NA	**NR**
Zhao J, et al. [24]	2022	France	French	Control DYX	31 26	13:18 13:13	12 ± 1:11 ± 2 11 ± 1:12 ± 1	NR	**Verbal (WISC);** **Non-verbal (WISC)**	**Word Reading** **Ability (Odedys);** **Word Spelling-to-** **Dictation Test**	**Nonword Reading** **Ability (Odedys)**	**Alouette** **Test**	**Digits; Objects**	Phoneme deletion and Spoonerism	NR	**VAS (Global** **and Partial** **Letter Report Task)**	**Verbal WM** **(DS- WISC)**
Meisler SL, [25]	2022	USA	English	Control DYX	582 104	195:387 44:60	10.8 ± 3.2 10.2 ± 2.5	NR	Non-verbal (KBIT-2)	SWE (TOWRE-2)	PDE (TOWRE-2)	NR	NR	NR	NR	NR	NR
Liu T, et al. [26]	2022	France	French	Control DYX	31 16	13:18 13:13	11 ± 1 12 ± 1	NR	**Verbal and Non-** **verbal (WISC-IV)**	NR; Global and Partial Letter Report Task (VAS)	NR	NR	**Digits; Objects**	**Phoneme deletion** **and Spoonerism**	NR	**NR**	**Verbal WM (DS- WISC)**
Farah R, et al. [27]	2022	USA	English	Control RD	24 22	12:12 10:12	8-12	NR	Non-verbal (TONI); Verbal (PPVT)	**SWE (TOWRE);** **LWID (WJ-III)**	**PDE (TOWRE);** **WA (WJ-III)**	NR	NR	**Elision** **(CTOPP)**	NR	**Conners questionnaires and VSA (TEA-Ch)**	**DS (WISC); Switching/Inhibition (DKEFS, Color-Word Condition); and Overall EF (BRIEF)**
Partanen M, et al. [28]	2020	Canada	English	Control DYX	22 13	11:11 5:8	Pre-test 8.5 ± 0.4 8.6 ± 0.4 Post-test 8.9 ± 0.4 8.9 ± 0.4	3rd Grade	Non-verbal (TONI-4)	Word Recognition Task (KTEA-II)	**Decoding Task** **(KTEA-II)**	Reading Comprehension (KTEA-II)	NR	NR	NR	NR	**NR**
Lou C, et al. [29]	2020	Canada	English	RD random group	64	33:31:00	10.9 ± 1.3	NR	NR	SWE (TOWRE)	PDE (TOWRE)	Reading Comprehension (WJ III)	Letters	NR	NR	NR	NR
Liu T, et al. [30]	2021	France	French	Control DYX	31 26	13:18 13:13	11 ± 1 12 ± 1	NR	**Verbal and** **Non-verbal (WISC)**	**Word Reading Fluency (Odedys);** **Word Spelling-to-Dictation Test**	**Nonword Reading** **Fluency (Odedys)**	**Alouette Test**	NR	NR	NR	NR	NR
Koirala N, et al. [31]	2021	USA	English	Random group	244	151:34:00	10.2 ± 2.8	NR	FSIQ(WISC)	**SWE (TOWRE-2)**	**PDE (TOWRE-2)**	NR	NR	**Elision and** **Blending Words** **(CTOPP-2)**	NR	NR	NR
Huber E, et al. [32]	2021	USA	English	Control DYX	41 32	16:25 12:20	9.4 9.8	NR	NR	**SWE (TOWRE)**	**PDE (TOWRE)**	**WJ-RF**	NR	NR	NR	NR	**WJ-MFF; WJ-CALC; and WJ-BRS**
Borghesani V, et al. [33]	2021	USA	English	Control DYX	14 26	5:9 14:12	10.4 ± 1.6 10.4 ± 2.0	1st grade and 4th grade	Non-verbal (WASI)	**SWE (TOWRE-2)**	**PDE (TOWRE-2)**	Gray Oral Reading Test (GORT-5)	NR	NR	NR	NR	NA
Vander Stappen C, et al. [34]	2020	France	French	Control DYX	13 18	5:8 9:9	10.5 ± 0.8 10.6 ± 1.0	NR	Non-verbal (WISC-IV)	**SWE - BALE**	**BALE**	**BALE**	**Objects; Colors**	**Syllable and** **phoneme deletion task**	NR	NR	NR
El-Sady S, et al. [35]	2020	Egypt	Arabic	DYX	20	05:15	8.2 ± 1	NR	SB4	1 min reading DAT	nonsense passage reading DAT	1 min reading DAT	Objects	Phonemic segmentation subtest of DAT	NR	NR	Bead threading; Postural stability; and DS
Wang HLS, et al. [36]	2019	Taiwan	Mandarin Chinese	Control DYX	22 24	NR	9 ± 0.9 10 ± 1	Primary school	Non-verbal (WISC-IV)	Chinese character recognition	NR	NR	NR	**NR**	NR	NR	**Lexical tone** **awareness;** **auditory** **identification** **of FM test**
Vanderauwera J, et al. [37]	2019	Netherlands	Dutch	TR and RD	34	19:15	13.7 ± 0.5	grade 8 (28), 7 (3) and 9 (3)	WISC-III-NL	One-minute word reading test	Klepel test	NR	NA	NA	NA	NA	NR
Lou C, et al. [38]	2019	France	French	Control DYX	31 26	13:18 13:13	11.5 ± 1.4 11.6 ± 1.3	NR	**Verbal and** **Non-verbal (WISC)**	**Word reading** **test (Odedys)**	**Nonword** **reading test** **(Odedys)**	Alouette test	**Digits;** **Objects**	**Phoneme deletion** **and spoonerism**	**Word spelling-** **to-dictation test**	NR	**Verbal WM** **(DS- WISC)**
Lebel C, et al. [39]	2019	USA	English	Dysfluent inaccurate Dysfluent accurate Non-impaired	20 36 14	5:15 13:23 8:6	10.0 ± 1.2 9.4 ± 1.3 9.2 ± 1.2	NR	**WASI FSIQ**	**SWE (TOWRE);** **LWID (WJ)**	**PDE (TOWRE);** **WA (WJ)**	**Gray Oral** **ReadingTest** **(GORT-4)**	NR	**Phonological** **Decoding** **(TOWRE)**	NR	NR	NR
Banfi C, et al. [40]	2018	Austria	German	TR DYX SD	27 21 21	12:15 9:12 6:15	9 ± 0.1 9 ± 0.3 10 ± 0.6	The end of 3rd and 4th grade	Verbal and Non-verbal (WISC)	**(SRLT-II);** **Spelling (DRT-3)**	**(SRLT-II)**	**Sentence** ** reading** **fluency(SLS)**	**Digits;** **Objects**	**PA**	Vocabulary standard score (WISC-IV)	**Parental ** **questionnaire ADHD**	Verbal WM and processing speed (DS, Symbol search WISC-IV)
Žarić G, et al. [41]	2018	Netherlands	Dutch	TR DYX	13 15	8:5 7:8	9 ± 0.8 9 ± 0.6	2–3 years of reading instruction	Non-verbal (WISC)	**Word reading** **subtest (3DM)**	**Pseudoword** **reading subtest (3DM)**	NR	**Letters;** **Digits;** **Objects**	**Phoneme deletion; Spelling;** **and Letter speech** **sound matching**	NR	NR	Memory span (syllables)
Su M, et al. [43]	2018	China	Mandarin Chinese	Control DYX	22 18	11:11 7:11	11 ± 0.8 11 ± 1.0	Primary school	Non-verbal and Verbal (C-WISC)	**Word list** **reading; Chinese** **character recognition**	NR	NR	**Digits**	**Phoneme deletion**	**Lexical decision;** **Morphological** **production**	NR	**Verbal WM** **(Digit recall)**
Yagle K, et al. [42]	2017	USA	English	TR DYG DYX	10 9 10	NR	9–14	4–9 grades	Non-Verbal (Wechsler)	**Word reading** **(TOSWRF); word** **spelling (TOC)**	Nonword reading	NR	NR	NR	NR	NR	NR
Christodoulou JA, et al. [44]	2016	USA	English	TR DYX	26 26	NR	7.8 ± 0.6 7.8 ± 0.6	NR	Non-verbal (KBIT-2)	**WID (WRMT-III);** **SWE (TOWRE-2)**	**WA (WRMT-III);** **PDE (TOWRE-2)**	NR	NR	NR	NR	NR	NR
Zhao JT, et al. [45]	2016	France	French	Control DYX	31 26	13:18 13:13	11.5 ± 1.3 11.6 ± 1.3	NR	**Verbal and** **Non-verbal (WISC)**	**Word reading** **fluency (Odedys);** ** Word spelling**	**Nonword reading** **fluency (Odedys)**	Alouette Test	**Digits, Objects**	**Word spelling-** **to-dictation test, Spoonerism**	NR	NR	**DS (WISC)**
Koerte IK, et al. [46]	2015	Germany	German	Control DYX	24 16	0:24 0:16	9.9 ± 0.3 9.7 ± 0.4	3rd and 4th grades	Non-verbal (CFT-20R)	**SLRT-II**	**SLRT-II**	**NR**	**Digits, Letters,** **Colors, Objects**	**Phoneme** **deletion**	NR	NR	DS (K-ABC); Verbal WM (Wechsler); Arithmetic test (HRT 1-4); and Number line task (WRT 1-4)
Garcia-Zapirain BG, et al. [47]	2016	Spain	Spanish	TR DYX MVR	19 20 18	8:11 8:12 8:10	10.0 ± 0.9 10.5 ± 1.1 10.4 ± 0.9	NR	**Verbal and** **Non-verbal (WISC-IV)**	**Word reading (PROLEC-R)**	**Pseudoword** **reading (PROLEC-R)**	ELFE 1-6	NR	NR	NR	NR	WM
Fernandez VG, et al. [48]	2016	USA	English	TR DYX	27 29	15:12 14:15	10.1 ± 2.1 12.1 ± 2.5	6–8 grades	**Verbal and** **non-verbal (KBIT-2, SB4)**	**LWI (WJ-III);** ** WRAT-3; and** **SWE (TOWRE)**	**PDE (TOWRE)**	PC (WJ-III-TA)	NR	NR	NR	NR	NR
De Moura LM, et al. [49]	2016	Brazil	Portuguese	TR RD	23 17	12:11 9:8	9.7 ± 0.9 9.2 ± 0.9	NR	Verbal and Non-verbal (WISC-III)	**Aloud reading** **(TDE)**	NR	NR	NR	NR	NR	NR	NR
Richards TL, et al. [50]	2015	USA	English	Control DYX DYG	9 17 14	5:4 7:10 3:11	mean of 12.25 (from 9 to 15.6)	4–9 grades	**Verbal** **(WISC-IV)**	**Spelling dictated** ** words (WIAT III);** **Sight Spelling (TOC)**	NR	NR	NR	NR	NR	NR	**Best and Fast** **writing (DASH)**
Marino C, et al. [51]	2014	Italy	Italian	TR FRD+ TR FRD− DYX FRD+ DYX FRD−	10 16 11 10	5:5 6:10 6:5 4:6	19.1 ± 1.9 18.7 ± 2.4 17.5 ± 2.4 16.4 ± 1.0	12.8 ± 1.5 12.0 ± 1.1 10.2 ± 1.8 10.8 ± 1.2	Full-scale IQ (WISC-R)	**Word reading(BVDDE);** **Spelling (BVDDE)**	**Non-word** **reading (BVDDE)**	**Sentences** **containing** **homophones**	NR	**Spoonerism,** **phonemic blending, and** **syllable displacement (PA)**	NR	NR	**ADC, letter and** **number forward/backward** **span (TEMA)**
Fan Q, et al. [52]	2014	USA	English	Control DYX	20 19	9:11 8:11	12.0 ± 0.7 12.0 ± 0.7	NR	**Verbal and** **Non-verbal** **(WISC-IV)**	**LWID (WJ-III);** **SWE (TOWRE); and** **FLI and spelling** **(WIST)**	**WA (WJ-III);** **PDE (TOWRE)**	**PC and basic** **reading (WJ-III);** **TOSCRF**	**Color Digit** ** Objects (CTOPP)**	NR	NR	NR	NR
Fan Q, et al. [53]	2014	USA	English	TR RD	16 20	8:8 8:12	11.7 ± 0.7 12.1 ± 0.7	NR	**Verbal and** ** Non-verbal** **(WISC-IV)**	**LWID (WJ-III);** **SWE (TOWRE);** **and FLI (WIST)**	**WA (WJ-III);** **PDE (TOWRE)**	NR	NR	WJ-III-PC	NR	NR	NR
Hasan KM, et al. [54]	2012	USA	English	TR DYX CFP	11 24 15	3:8 11:1 39:6	12.8 ± 1.7 13.7 ± 1.0 13.5 ± 0.8	NR	**Composite IQ** **(KBIT-2, SB4)**	**LWID (WJ-III);** **SWE (TOWRE)**	**PDE (TOWRE)**	**PC (WJ-III)**	NR	NR	NR	NR	NR
Gebauer D, et al. [55]	2012	Austria	German	Control SI SRI	11 11 9	NR	12.3 ± 1.9 11.7 ± 1.6 11.3 ± 0.7	4th–5th 5–9 graders	Non-verbal (Raven)	**SLS 1-4 or 5-8;** **Spelling (HSP)**	**SLS 1-4** **or 5-8**	**ELFE 1-6**	NR	NR	NR	NR	Personality assessment FFQ (Asendropf)
Hoeft F, et al. [56]	2011	USA	English	Control DYX (rg) DYX (nrg)	20 13 12	14:6 6:7 7:5	11.0 ± 2.6 14.5 ± 1.6 13.5 ± 2.2	NR	**Non-verbal** **(WASI)**	**(WRMT) *;** **SWE (TOWRE);** **and Spelling and** **writing fluency (WJ)**	**WA (WRMT);** **PDE (TOWRE)**	**Gray Oral** **ReadingTest(GORT);** **PC (WRMT)**	**Colors; Objects;** **Numbers; and Letters**	NR	**PPVT**	NR	**MD (CTOPP)**
Sihvonen AJ, et al. [57]	2021	Finland	Finnish	Control DYX	21 23	11:10 12:11	29.9 ± 6.0 31.3 ± 8.6	16.1 ± 4.4 15.7 ± 5.2	**Verbal (WAIS-III);** **PIQ (WAIS-IV)**	**Word List** **Reading**	**Pseudoword** **List Reading**	**Text Reading**	**Test not** **specified**	**Pig Latin;** **PA; phonological** **short-term memory;** ** and rapid access** **of information**	NR	ASRS v1.1	ARHQ;Verbal WM (Non-word Span Length, WMS-III)
Tschentscher N, et al. [58]	2019	Germany	German	Control DYX	12 12	0:12 0:12	23.7 ± 2.6 24.2 ± 2.3	NR	Nonverbal (Raven)	**Spelling**	NR	**Reading speed** **and** **comprehension**	**Letters and** **Numbers**	NR	NR	NR	NR
Moreau D, et al. [59]	2018	New Zealand	English	Control Dyscalc DYX Comorbid	11 11 11 12	4:7 5:6 5:6 5:7	27.7 ± 1.7 32 ± 2.2 29.4 ± 1.9 33.2 ± 1.7	15.2 ± 0.60 14.6 ± 0.56 15.6 ± 0.51 14.8 ± 0.61	FSIQ (WASI)	WID (WJ)	WA (WJ)	NR	NR	NR	WRAT spelling	NR	WRAT mathematics
Müller-Axt C, et al. [60]	2017	Germany	German	Control DYX	12 12	0:12 0:12	23.7 ± 2.6 24.2 ± 2.4	Undergraduate students **	Non-verbal (Raven)	**Spelling**	NR	NR	**Numbers;** **Letters**	NR	NR	NR	NR
Vandermosten M, et al. [61]	2013	Belgium	Dutch	TR DYX	20 20	12:8 13:7	21.4 ± 3.0 22.1 ± 3.1	NR	Non-verbal (WAIS-III)	**Word reading;** **Spelling**	**Pseudoword** **reading**	NR	NR	NR	NR	NR	NR
Lebel C, et al. [62]	2013	USA	English	RLD	136	64:13:00	20.1 ± 3.1	NR	FSIQ (WASI)	**WID (WJ)**	**WA (WJ)**	**Fluency** **(GORT)**	NR	NR	NR	NR	NR
Vandermosten M, et al. [63]	2012	Belgium	Dutch	TR DYX	20 20	12:8 13:7	21.4 ± 3.0 22.1 ± 3.1	NR	Non-verbal (WAIS)	**Word reading;** **Spelling**	**Pseudoword** **reading**	NR	NR	**PA; Phoneme** **deletion and** **Spoonerism**	Speech-in- noise perception (Dutch LIST)	NR	NR
Frye RE, et al. [64]	2011	USA	English	TR DYX/PR	20 10	10:10 5:5	23.7 ± 0.7 23.9 ± 1.6	NR	**Non-verbal** **(CTONI)**	**LWI (WJ-III);** **Spelling**	**WA (WJ-III)**	**Gray Oral** **ReadingTest** **(GORT)**	**Colors (CTOPP);** ** Digits (CTOPP); ** **Objects (CTOPP);** **Letters (CTOPP)**	**PA (CTOPP);** **APA (CTOPP)**	NR	Test of variables of attention: commissions, omissions	NR

Note: Bold font indicates significant group differences. * time effect in DYX group, ** only 1 control had a high school diploma. Abbreviations: NA: Not Applicable; NR: Not Reported;
FRD+: Children with Familial Risk for Dyslexia; FRD−: Children without Familial Risk for Dyslexia; TR: Typical reading; rg: reading gain; nrg: no rg; m: months; RAN: rapid
automatized naming tasks; DYX: children with dyslexia; RD: Reading Disorder; RI: Reading Impairment; PreR: pre-reader children; BR: older reader children; FR: fluent reader children;
PIQ: Performance IQ; FSIQ: Full-Scale Intelligence Quotient; CFT 20-R: Cattell’s Fluid Intelligence Test, Scale 2; TrR: Treatment Responders; EF: Executive Functions; MD: Mood
Disorders; WRD: word recognition deficits; RLD: reading and learning disabilities; SI: spelling impaired children; SRI: children with spelling and reading impairment; CFP: readers with
comprehension or fluency problems; PA: Phonological awareness; PPVT: Peabody Picture Vocabulary Test; TONI or TONI-4: Test of Nonverbal Intelligence; CTONI: Comprehensive
TONI; CTOPP: Comprehensive Test of Phonological Processing; TOWRE or TOWRE-2: Test of Word Reading Efficiency; WJ: the Reading Fluency subtest of the Woodcock-Johnson Test;
WISC or WISC-IV: Wechsler Intelligence Scale for Children, 4th edition; WASI: Wechsler Abbreviated Scale of Intelligence; WAIS: Wechsler Adult Intelligence Scale, 3rd edition; WIAT:
Wechsler Individual Achievement Test; BRIEF: Behaviour Rating Inventory of Executive Function; KBIT or KBIT-2: Kaufman Brief Intelligence Test; WRMT-R NU: Woodcock Reading
Mastery Tests-Revised, Normative Update; WA: Word attention; SLRT-II: The Salzburg Reading and Spelling Test; YARC: York Assessment of Reading for Comprehension; GAPS:
Grammar and Phonology Screening; CELF or CELF-4: Clinical Evaluation of Language Fundamentals; KTEA-II: Kaufman Test of Educational Achievement-Second Edition; CVLT:
California Verbal Learning Test; WID: word identification; LWID: letter and WID; VAS: Visual Attention Span; SB4: Stanford-Binet Intelligence Scales-Fouth Edition; HLE: Home Literacy
Environment; HOME: the Home Observation for Measurement of the Environment; DAT: Dyslexia Assessment Test; ASRS: Adult Self Report Scale for ADHD clinical assessment; TOC:
Test of Orthographic Competence; TOMAL2: Test of Memory and Learning - Second Edition; MSEL: Mullen Scales of Early Learning; ARHQ: Adult Reading History Questionnaire;
ADC: Adult Dyslexia Checklist; HSP: Hamburger-Schreibprobe; SLS: Salzburger-Lese-Sreening; FFQ: five factor questionnaire; DAWBA: Diagnostic andWell-Being Assessment; SWE:
SightWord Efficiency; DS: digit span; VSA: visual spatial attention; BISC: Bielefeld screening of literacy precursor abilities; DERET: German spelling test; SETK 3 5: a developmental
German language test for children between 3 and 5 years of age; BAKO: Test of basic reading and spelling skills; PDE: Phonemic Decoding Efficiency; TOSWRF or TOSWRF-2: Test
of Silent Word Reading Fluency, Second Edition; GORT: Grey Oral Reading Test; PC: Passage comprehension; ODEDYS: dyslexia screening tool; DKEFS: Delis–Kaplan Executive
Function System; BALE: Analytic Battery of Written Language; DRT-3: Spelling percentile; 3DM: differential diagnostics for dyslexia; HRT: Heidelberger Rechentest; WRT: Weingartener
Grundwortschatz Rechtschreibtest; PROLEC: Text Comprehension task; WRAT: Wide Range Achievement Test; TDE: test for School Achievement; DASH: Detailed Assessment of Speed
of Handwriting; FLI: Fundamental Literacy Index; WIST: Word Identification and Spelling Test; ELFE: standardized achievement tests; TEMA: Test di Memoria e Apprendimento;
BVDDE: Battery for the Assessment of Developmental Reading and Spelling Disabilities; and WM: working memory.

**Table 2 brainsci-14-00349-t002:** DTI acquisition, image processing, and outcomes.

	**DTI Acquisition**									**DTI Processing**						**DTI Outcomes**	
**Ref**	**MRI Field**	**Sequence**	TR/TE (ms)	Slice Number	Slice Thickness (mm)	**FOV (mm)**	b-Value (s/mm^2^)	N. of Diffusion Gradients	Time	Software	Corrections	Type of Analyses	DTI Metrics	Atlas	ROIs/ TRACTS	Tracts Difference between Groups	Clinical Correlations
Zuk J, et al. [16]	Siemens 3T	DTI	NR	30	2	128x128	0; 700	NR	NR	DTIprep, VISTALab	EC, HM (>2 mm or >0.5°)	ROI	FA	NR	AF, SLF	↑FA in r-SLF of FHD+ TR compared to FHD− TR and FHD+ RD	r-SLF FA, age, gender, parent education, occupation, and phonological awareness significantly predicted decoding skills among children FHD+
Yu X, et al. [17]	Siemens 3T	DTI	NR	NR	2	NR	0; 700; 1000	NR	NR	DTIprep, VISTALab, AFQ	EC, HM (>2 mm/0.5°), bed vibration, pulsation, venetian blind artifacts, and slice and gradient-wise intensity inconsistencies	Whole brain; ROI	FA	MNI, Native space	Right of SLF, ILF, and AF, sCC, CC2	↑FA in r-sCC of FHD+ TR compared to FHD− TR	R-sCC FA had positive correlation with r-IFG activation for FHD−/+ TR
Langer N, et al. [18]	Siemens 3T	DTI	8320/88	64	2	256x256	1000	30	5:59 min	DTIprep, FSL (DTIFIT), Trackvis (Diffusion Toolkit), Trackvis, AFQ	EC and HM (>2 mm and 0.5°)	Whole brain; ROI	FA RD AD	MNI	Bilateral AF and CS	↓FA in l-AF (central portion) FHD+ compared with FHD−, corrected by age	l-AF FA has positive correlation with age, expressive language
Kraft I, et al. [19]	Siemens 3T	DTI	8000/NR	66	1.9	NR	1000	60	32 min	FSL (Topup tool), FSL (DTIFIT), MRTrix	EC, HM, and susceptibility- induced distortions	ROI	FA	Destrieux Atlas	SMG, ITG (anterior, long, and posterior AF), SOS/TOS, IFoG, IFobG	No group difference	l-aAF was the best predictor of DYX
Vanderm- osten M, et al. [20]	Philips 3T	DTI	7600/65	58	2.5	200x240	1300	60	10 min 32 s	Explore DTI, Trackvis	EC, HM (6 parameters) Reorientation of the b-matrix Motion as covariate	Whole brain; ROI	FA	TrackVis	AF (dorsal FTP, dorsal post TP), ventral IFOF	↓FA in l-IFOF of FHD+	Phonological awareness positive correlation with FA of l-AF(TP) and bilateral IFOF/AF-FTP, as also left ventral tracts in FHD+
Van Der Auwera S, et al. [21]	Philips 3T	DTI	7600/65	NR	2.5	NR	1300	60	10:32 min	FSL, VISTALab, AFQ	EC, HM by root mean square	Whole brain; ROI	FA MD	NR	AF	↓FA in the l-AF in pre-reader RDs	aAF FA was a significant predictor for scores on word reading tests from 2nd grade
Wang Y, et al. [22]	Siemens 3T	DTI	8320/88	NR	NR	256x256	1000	30	5:59 min	DTIprep, VISTALab, AFQ	EC, HM (>2 mm and >0.5°)	Whole brain; ROI	FA AD RD	white matter atlas	Left of AF, SLF, ILF	↓l-AF FA at pre-reader FHD+ versus FHD− and for poor versus good readers all ages; FHD+ good readers had faster WM development in r-SLF compared to poor readers	l-AF and ILF FA positive correlations with word identification skill
Vanderau- wera J, et al. [23]	Philips 3T	DTI	7600/65	NR	2.5	NR	1300	60	10:32 min	ExploreDTI, Trackvis	EC, HM	ROI	FA	native space	Long, anterior and posterior dorsal AF, and ventral IFOF	↑FA in all groups over time. ↓ long AF FA in DYX prior to reading onset, right also kept in early reading. Influence of FHD+ in l-IFOF and long r-AF	FHD+ and rapid naming predicted 80.3% of cases; the l-longAF FA values predicted 84.4% of DYX cases
Zhao J. et al. [24]	Siemens 3T	DTI	14,000/91	70	1.7	218	1400	60	18 min	Explore DTI, Trackvis, FSL	NR	Whole brain	FA	TrackVis MNI-152	UF, FAT	Males DYX had a ↓HMOA in the UF compared with males TR	HMOA of the UF showed a positive correlation with VAS in DYXs
Meisler SL, [25]	Siemens 3T	DKI	3320/100.2	NR	1.8	NR	0; 1000; 2000	64	NR	QSIPrep, MRtrix, FSL, and TractSeg	Gibbs unringing, EC, HM, and AP-PA field	Whole brain	FA	FSL and MNI	AF, SLF (I, II, and III), ILF, IFOF, UF, SCP, ICP, MCP, and sCC	No group difference	Age and sex with gFA positive correlation; in older children, FA in r-SLF and l-ICP related to nonword reading skills
Liu T, et al. [26]	Siemens 3T	DTI	14,000/91	70	1.7	218	1400	60	18 min	PANDA, FSL, and Trackvis	EC	Whole brain	FA	MNI and AAL atlas	90 ROIs of AAL	NR	Positive correlation between node FA for l-SOG and VAS score, l-MTG and l-ORBsupmed and phonological score
Farah R, et al. [27]	Philips 3T	DTI	6652.446/82.6	160	2	224x120x224	1000	61	7 min 25 s	VISTALab, AFQ	EC, HM	Whole brain; ROI	FA	NR	AF, SLF, ILF	↓ FA in the left of AF, ILF, and SLF in RD	↓ FA in the l-SLF positive correlated with reading and working memory score in DYX
Partanen M, et al. [28]	GE 3T	DTI	7000/60	60	2	256x256	0; 1000	60	7.5 min	TORTOISE, FDT (FSL), DTIFIT (FSL), and PROBTRACKX (FSL)	EC, HM	Whole brain; ROI	FA MD	MNI305 and Desikan–Killiany atlas	bilateral IFG, Ins, STG, SMG, AnG, and FFG	↑MD in bilateral Ins; l-IFtG, l-STG, and r-SMG in DYX	SMG, r-IFoG, and l-Ins MD had negative correlation with reading gains and decoding, respectively
Lou C, et al. [29]	Siemens 3T	DTI	3000/50.6	64	2	256x256	0; 1000	56	NR	ExploreDTI	EC, HM, EPI distortions	Whole brain; ROI	FA	AAL and MNI152	90 ROIs of AAL	NR	IFtG and IFoG, Ins, FFG, IPL, SMG, AnG, HG, STG, MTG, ITG, IOG, PreCG, ROL, and thalamus in the left hemisphere positive correlated with reading efficiency and phonemic decoding, mainly for girls DYX
Liu T, et al. [30]	Siemens 3T	DTI	14,000/91	70	1.7	218x218	0; 1400	60	18 min	PANDA, FSL	EC, HM	Whole brain; ROI	FA	AAL and MNI	90 ROIs of AAL	NR	Negative correlation between READACC (pseudoword/word reading) and the r-FFG FA in DYX
Koirala N, et al. [31]	Siemens 3T	DTI	NR	NR	1.8	NR	0; 1000; 2000	64	NR	FSL (QUAD), FSL (DTIFIT), and FSL (BEDPOSTX), XTRACT	Susceptibility, EC, and HM	Whole brain; ROI	FA MD RD ODI NDI	Native space	23 tracts (including SLF, which seeds were central sulcus, SFG, ACG, MFG, and AnG)	NR	Positive correlation between phonological processing and the left IFOF, MDLF, SLF2, VOF, CBD and FX FA, and the l- UF MD
Huber E, et al. [32]	Phillips 3T	DKI	NR	NR	2	NR	0; 800; 2000	32 and 64	NR	FSL, DIPY, MRTrix, and AFQ	AP-PA, EC, Mean slice-by-slice displacement > 3 mm	Whole brain	FA MD AWF Da MDe	NR	AF, CS, UF, SLF, ILF, ThR, FMj, FMn, and IFOF	l-AF MD difference for Group x time interaction	Positive correlation between MD of l-AF, UF, l-ILF, l-IFOF, FMj, MDe of left of AF, UF, ILF, IFOF, and FMj with word reading and negative correlation between AWF of right ILF, IFOF, and FMn with word reading
Borghesani V, et al. [33]	Siemens 3T	DKI	8200/86	60	2.2	220x220	0; 700; 20,000	30 and 64	15 min	FSL (NODDI model), FS-TRACULA	AP-PA, EC, and HM	Voxel-based; ROI	NDI ODI	FSL, Desikan–Killiany Atlas and MNI	l-VOT	↑ODI in DYS at the l-VOT	NR
Vander Stappen C, et al. [34]	Philips 3T	DTI	6422/83	70	2	224x224	800	55	NR	BrainVoyager	EC, HM	ROI	FA	Talairach space	AF, IFOF, and ILF	NR	RAN Gains negative correlated with FA in the l-long aAF, and the r-pAF, a reduction in naming times was linked to an increase in FA in those tracts at DYX
El-Sady S, et al. [35]	Philips 1.5T	DTI	NR	70	2	230x230	NR	32	NR	NR	EC, HM	ROI	FA ADC	NR	SLF, AF, CR, PLIC of CS	NR	Negative correlation between r-AF FA and at-risk quotient, l-sCR ADC with writing, and r-SLF ADC with bDS and positive with VF.Positive correlation between l-SLF-aCR FA and RAN, spelling, and VF, as r-PLIC ADC with writhing, and l-aCR ADC with bDS
Wang HLS, et al. [36]	Siemens 3T	DTI	6700/97	NR	2.7	NR	5000	128	NR	DSI Studio	NR	Whole brain	NR	MNI, AAL atlas	IFOF, CC, cerebellar, and Tha-pontine tracts	NR	l-IFOF, cerebellar, and Tha-pontine tracts had positive correlated with chinese character recognition; pCC association with auditory FM processing in DD
Vanderauwera J, et al, [37]	Phillips 3T	DTI	8872/2.5	55	2.5	240x240x137.5	1000	60	13:52 min	ExploreDTI, Trackvis	HM (>1.5 mm) and EC	ROI	FA	Native Space	AF, IFOF, UF, and ILF	NR	Word reading had positive correlation with l-long-AF FA and negative with l-long-AF RD and UF RD. Paternal educational level had positive correlation with l-long AF FA, and UF FA; after covariate by HM, only the l-UF remained significant
Lou C, et al. [38]	Siemens 3T	DTI	14,000/91	70	1.7	218	0; 1400	60	18 min (3x6 min)	ExploreDTI, FSL (FLIRT)	EC, HM	Whole brain	FA	AAL atlas; MNI; Harvard-Oxford atlas	Left of MTG-MOG, MOG-TPOsup, TPOsup-HG, HG-ROL, Ins-ROL, STG-Ins, and Ins-SMG	↓mean FA in DYX for all ROIs	Literacy skills had positive correlation with clustering coefficient, local efficiency, transitivity, and global efficiency, in DYX
Lebel C, et al. [39]	Siemens 1.5T	SE EPI	9000/85	28	5	240x240	1000	NR	7:24 min	FSL	Motion artifacts (signal drop out, venetian blind artifact, and mechanical vibration artifact, >10), EC	ROI	FA MD AD RD	MNI; JHU ICBM- DTI-81 atlas	sCC, ALIC of CS, aCR, pCR, SS (includes the ILF and IFOF), UF, and SLF	↓MD in r-CR, and l-UF in DYX	Age had a positive correlation with pCR, r-SLF FA, negative with pCR, l-UF MD. Sight words and VF were positively correlated with l-SLF FA and MD, respectively, as well as with l-pCR MD. Phonological decoding had a negative correlation with r-pCR MD and mean MD and positive with mean FA
Banfi C, et al. [40]	Siemens 3T	DTI	3400/105	48	2.5	240	0; 2000	64	NR	MRTrix, FSL, and AFQ	AP-PA, EC, HM, and susceptibility- induced distortion	Whole brain	FA	NR	ThR, FMj, FMn, IFOF, ILF, SLF, and AF, UF, CS, and CG	↑FA in ILF, r-SLF, and r-CG in DYX	Negative correlation between r-ILF FA and reading measures, controlling for spelling.
Žarić G, et al. [41]	Siemens 3T	DTI	10,800/84(protocol1) 11,000/85(protocol2)	85	1.8	NR	0; 1000	72	15 min	VISTALab (mrDiffusion), SPM, AFQ	EC, HM and phase-encoding direction corrections	Whole brain; ROI	FA	NR	AF, SLF, ILF, IFOF, UF, lCS, antThR, FMn, and FMj	↑FA in the AF, r-SLF, and aThR in DYX	r-SLF showed age effects that differed between groups. Age effect in ILF FA, and CC (FMj and FMn). L-aThR positive correlation with age appropriate reading accuracy scores
Su M, Zhao J, et al. [43]	Siemens 3T	DTI	8000/89	NR	2.2	282x282	0; 1000	30	NR (repeated twice)	ExploreDTI, Trackvis, and FSL	EC and HM	Whole brain; ROI	FA RD AD	MNI152; native space	AF, IFOF, and ILF	↓FA and AD in the l-AF and I-ILF in DYX	AF and ILF FA positive correlation with character recognition, digit recall, phoneme deletion (only AF), and morphological production (only ILF). ILF FA negative correlation with RAN
Yagle K, et al. [42]	NR	DTI	8593/78	NR	2	220x220x128	0; 1000	32	9:35 min	FSL	NR	ROI	FA RA AD RD MD	NR	OR, CS, ILF, SLF, and CG	↓FA in l-OR in DYX	NR
Christod- oulou JA, et al. [44]	Siemens 3T	DTI	9300/84	74	2	256	0; 700	30	NR	FS-TRACULA, DTIprep, and FSL (FLIRT)	EC, HM	Tract-based	FA AD RD	MNI152	SLF, AF	↓FA in the l-AF in RDs	Positive correlation of l-AF FA and negative DA with real-word reading
Zhao JT, et al. [45]	Siemens 3T	DTI	14,000/91	70	1.7	218	0; 1400	60	18 min	ExploreDTI, FSL, and Trackvis	Motion corrections	Whole brain; ROI	HMOA FA	MNI152	IFOF, ILF, SLF, and AF	↓FA of r-IFOF and l-SLF in DYX	r-IFOF FA negative correlation with reading and spelling accuracy
Koerte IK, et al. [46]	Siemens 3T	NR	9600/110	65	2	208	0; 1000	30	NR	3DSlicer, FSL (FLIRT), and FSL (TBSS)	EC, HM	Tract-based	FA AD RD trace	MNI152	NR	No group difference	Positive correlation arithmetic test with FA and AD and negative with RD (Temporo-parietal)
Garcia-Zapirain BG, et al. [47]	Philips 3T	DTI	6819/81	60	2	224x224	800	15	7min	FSL (BET), FSL (FDT), and FSL (TBSS)	NR	Whole-brain; ROI	FA MD AD RD	MNI; Atlas JHU White- matter	CC, SLF, ILF, lower FOF, l-AF, IFOF	↓FA in l-AF in DYX	NR
Fernandez VG, et al. [48]	Philips 3T	DTI	6100/84	44	3	240x240	0; 1000	21	NR	FSL (DTIFIT)	EC, HM	ROI	FA AD RD	Desikan and Destrieux atlases	LAC/RAC to bilateral TP, OT, and IFG	↑FA of cerebellar to TP and IFG; ↓RD in TP in DYX	FA of AC-OT had interaction between age and group, younger DYX have ↓FA in this region.
De Moura LM, et al. [49]	GE 1.5T	DTI	11,600/99	47	3	240x240	0; 800	15	NR	FSL, FSL(TBSS)	EC correction and non brain voxels removed	Voxel-based	FA RD MD AD	MNI152	aThR, CG, CS, IFOF, ILF, UF, FMj, FMn, and CGH	↓FA left of aThR, CG, CS, FMj, FMn, UF, right of IFOF, ILF ↑RD in the left of CG, CS, and SLF in DYX	NR
Richards TL, et al. [50]	Philips 3T	DTI	8593/78	NR	2.0	220x220x128	0; 1000	32	9 min 35 s	DTIPre (GTRAC), FSL (TBSS), and FSL	NR	ROI	FA AD RD RA MD	FSL white matter atlas (FHU)	aThR, FMn, CS, SLF, ILF, IFOF, UF, and CG	↓RA in aThR, IFOF, SLF, UF, and l-CG, and FMn; ↓AD in CS, r-ThR, CG, IFOF, SLF, and UF in DYX	NR
Marino C, et al. [51]	Philips 3T	DTI	9775/58	NR	2.3	NR	0; 1000	35	NR	BrainVoyager (Brainvisa), SPM	EC, smooth 6 mm	Voxel-based	FA	White matter atlases of FSL	ILF, IFOF, AF, SLF, CC, and OR	NR	DYX with DCDC2d gene x without found ↓FA in ILF and l-CC
Fan Q, et al. [52]	Philips 3T	DTI	6237/75	60	2.2	212x212	0; 700	32	3 min 32 s	FSL (FDT), FS-TRACULA	EC, HM	ROI	FA	Desikan–Killiany Atlas	Thalamus to OFC, MPFC, LPFC, SMC, PC, MTC, LTC, OCC, and Ins	↑FA of LPFC and SMC to ThR in DYX	Th-SMC showed negative correlation with basic reading score
Fan Q, et al. [53]	Philips 3T	DTI	6237/75	60	2.2	212x212	700	32	3 min 38s	FSL	EC, HM	ROI	NR	MNI152	5 ROIs of l-OT/F, MTG, ITG, LOCC, PaHipp, and ILF	Left Mid, Inf and sup- TG, lingual, fusiform, Sup and Inf PG in DYX	NR
Hasan KM, et al. [54]	Philips 3T	DTI	6100/84	44	3	NR	1000	21	min	NR	NR	ROI	FA MD AD RD Dav	NR	CC	↑mFA of CC in DYX	MD and AD correlation with age (CC2); MD positive correlated with Letter- Word ID test in CC5
Gebauer D, et al. [55]	Siemens 3T	DTI	6700/95	35	2.5	250	NR	NR	NR	FSL (TBSS, FDT, DTIFIT, and BET)	EC	Voxel-based	FA	JHU ICBM-DTI-81 White-Matter Labels	aCR, CC	↓FA in the l-aCR and aCC	NR
Hoeft F, et al. [56]	GE 3T	DTI	11,600/64.5	23	4	240	800	13	NR	SPM, DTIStudio, and ROQS	EC, HM	Whole-brain	FA	NR	SLF	NR	Positive correlation between r-SLF FA and single-word reading
Sihvonen AJ, et al. [57]	Siemens 3T	DTI	9000/80	70	2.5	240x240	0; 1000	64	NR	MRTrix, DSI Studio	Thermal noise with MP-PCA, Gibbs ringing correction	Whole brain	QA	MNI using (QSDR)	NR	↓QA in VOF, SLF, AF, CC, CSl-UF, and ThR; ↑QA in l-SLF, VOF, and CS in DYX	Reading skill positive association with l-CG and right fornix, and frontal corticopontine tracts and cerebellum
Tschentscher N, et al. [58]	Siemens 3T	DTI	12,900/100	88	1.7	220x220	0; 1000	60	6 min	FSL (FDT), FSL (PROBTRACKX), and FSL (BEDPOSTX)	Head motion corrections	Voxel-based; ROI	FA	MNI; Juelich histological; Harvard-Oxford atlases	A1, l-mPT, and MGB, IC	↓connectivity between l-mPT-MGB in DYX	Negative correlation of l- mPT-MGB with reading skills in TR
Moreau D, et al. [59]	Siemens 1.5T	DTI	6601/101	NR	3	230	0; 1000	30	NR	FSL (DTIFIT), FSL (FLIRT), and FSL (TBSS)	EC and motion corrections	Whole brain; Voxel-based	FA	MNI152	Bilateral CR and AF	No group difference	NR
Müller-Axt C, et al. [60]	Siemens 3T	DTI	12,900/100	88	1.7	220x220	0; 1000	60	6 min	FSL	Motion correction	ROI	FA	Talairach; MNI; Juelich Histological atlas	LGN, l-V1, V5/MT	↓LGN FA and between l-V5/MT-LGA in DYX	DYX showed negative correlation between l- V5/MT-LGN and name letters and numbers aloud time
Vanderm- osten M, et al. [61]	Philips 3T	DTI	11,043/55	68	2.2	220x220	0; 800	45	21 min 8 s	Explore DTI, FSL (CATNAP)	EC and motion- induced artifacts	Whole-brain; ROI	FA	Harvard-Oxford atlas in MNI space	Post STG, AF, sCC	NR	Positive correlation between coherence 20 Hz and FA of the STGp Lat and sCC in DYX and a negative in HC, without outliers
Lebel C, et al. [62]	Siemens 1.5T	DTI	9000/85	28	5	240x240	0; 1000	6	7 min 24 s	SPM	Smooth of 4 mm kernel	Voxel-based	FA MD	ICBM template	ALIC, sCC, ThR, CR, ILF, IFOF, anf aCR	NR	GORT fluency positive correleted with FA of aCC, sCC, right: aLimb, SLF, MCP, aCR, ILF, l-sCC, Th, IFOF; Word attack with FA of aCC, SLF, aLimb; l-Th, SLF, and r-IFOF
Vandermosten M, et al. [63]	Philips 3T	DTI	11,043/55	68	2.2	220x220	0; 800	45	21 min 8 s	Explore DTI, FSL (CATNAP)	EC, motion-induced artifacts correction	ROI	FA RD AD	Native space	AF, IFOF	↓FA of l-AF in DYX	Direct and l-aAF FA positive correlated with phoneme awareness, and speech perception, respectively, and l-IFOF with orthography
Frye RE, et al. [64]	Philips 3T	DTI	6100/84	44	3	240x240	1000	NR	7 min	SPM	Distortion correction, masking, and isotropic voxel interpolation	Whole-brain	FA AD RD Dav	ICBM	FTP, SLF, SFOF, IFOF, and CR	No group difference	Negative correlated: FA- word attack in SLF, SFOF, aCR, and pCR; Dav- word attack in SLF; and positive correlation: Dav and AD—word attack in SFOF

Abbreviations: Ref.: Reference; MRI: magnetic resonance imaging; N: number; DTI: diffusion tensor image; DKI: diffusion kurtosis imaging; NR: Not reported; TR: Time of repetition; TE: time of echo; FOV: field of view; b: diffusion weighting; l: left; r: right; UF: Uncinate fasciculus; FAT: frontal aslant tract; EC: Eddy Current; ICBM: International Consortium for Brain Mapping; SLF: Superior longitudinal fasciculus; SFOF: superior frontal–occipital fasciculus.; IFOF: inferior frontal–occipital fasciculus; ROQS: Reproducible Objective Quantification Scheme; CC: corpus callosum; CC5: posterior midbody of CC; CC2: genu of CC; sCC: splenium of CC; aCC: anterior CC; VOT: ventral occipitotemporal cortex; OT/F: occipitotemporal/fusiform; SMC: supramarginal cortex; IFG: inferior frontal gyrus; FTP: frontal-temporo-parietal regions; TP: temporo-parietal regions; FMj: forceps major; FMn: forceps minor; ThR: thalamic radiations; aThR: anterior ThR; CG: cingulum; CS: corticalspinal tract; AF: arcuate fasciculus; aAF: anterior AF; pAF: posterior AF; ILF: inferior longitudinal fasciculus; HMOA: Hindrance-modulated oriented anisotropy; VAS: Visual Attention Span; AFQ: Automated Fiber Quantification software; SCP: superior cerebellar peduncle; ICP: inferior cerebellar peduncle; MCP: middle cerebellar peduncle; FA: fractional anisotropy, gFA: global white matter fractional anisotropy; AAL: automated anatomical labeling; MNI: Montreal Neurological Institute; SOG: superior occipital gyrus; MTG: middle temporal gyrus; ORBsupmed: medial orbital superior frontal gyrus; TR: typical readers; RD: reading disorder; QSDR: q-space diffeomorphic reconstruction; QA: quantitative anisotropy; FDT: FMRIB’s Diffusion Toolbox; MD: mean diffusivity; STG: superior temporal gyrus; ITG: inferior temporal gyrus; SMG: supramarginal gyrus; DYX: dyslexia; AnG: angular gyrus; IPL: inferior parietal lobe; IOG: inferior occipital gyrus; PreCG: precentral gyrus; ROL: Rolandic operculum; FFG: fusiform gyrus; QUAD: Quality Assessment of dMRI; DTIfit: diffusion tensor modeling tool; MDLF: middle longitudinal fasciculus; VOF: ventral occipital fasciculus; CBD: dorsal cingulum; FX: fornix; AWF: axonal water fraction; Da: intra-axonal diffusivity; MDe: extra-axonal mean diffusivity; NDI: neurite density index; ODI: orientation dispersion index; FHD+: positive familial risk to develop dyslexia; FHD−: negative familial risk to develop dyslexia; CR: corona radiata; aCR: anterior CR; pCR: posterior CR; PLIC: posterior limb of internal capsule; FM: frequency modulation; A1: primary auditory cortex; mPT: planum temporale; MGB: medial geniculate body; IC: inferior colliculus; BEDPOSTX: Bayesian Estimation of Diffusion Parameters Obtained using Sampling Techniques; MOG: middle occipital gyrus; -TPOsup: temporal pole; HG: Heschl’s gyrus; AD: axial diffusivity; RD: radial diffusivity; ALIC: anterior limb of the internal capsule; SD: spelling disorder; TBSS: Tract-based spatial statistics; FLIRT: FMRIB linear image registration tool; RA: relative anisotropy; SLD: specific learning disability; WM: white matter; TP: temporoparietal; LGN: lateral geniculate nucleus; V1: primary visual cortex; V5/MT: middle temporal area; TRACULA: TRActs Constrained by UnderLying Anatomy; TP-AF: temporo-paietal portion of the AF; ASD: autism spectrum disorder; SOS/TOS: superior and transversal occipital sulci; BET: Brain Extraction Tool; LAC: left anterior cerebellum; RAC: right anterior cerebellum; HC: health control; JHU: Johns Hopkins School; OFC: orbitofrontal cortex; MPFC: medial prefrontal cortex; LPFC: lateral prefrontal cortex; PC: parietal cortex; MTC: medial temporal cortex; LTC: lateral temporal cortex; OCC: occipital cortex; Ins: insular cortex; LOCC: lateral OCC; PaHipp: parahippocampal regions; GORT: Gray Oral Reading Test; aLimb: anterior limb; SS: sagital stratum; IC: inferior colliculus; OR: Opptic radiation; IFoG: pars opercularis of inferior frontal gyrus; IFtG: pars triangularis of IFG; IFobG: pars orbitalis of IFG; bDS: backward digit span; and VF: verbal fluency.

## Data Availability

Not applicable.

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
