# Peer review of "Investigating Dyslexia through Diffusion Tensor Imaging across Ages: A Systematic Review"

_brainsci, 2024, doi:10.3390/brainsci14040349_

Round 1
Reviewer 1 Report
Comments and Suggestions for Authors
This structure literature review focuses on studies using structural neural imaging (specifically measures of white matter integrity) to compare/contrast participants with and without dyslexia. The methodology and presentation of results are appropriate. However, the manuscript would benefit from enhancements to the Introduction, clarification of some methods and results, and phrasing/grammatical corrections throughout. Please see below for specific comments.
Misc:
· Developmental dyslexia is a neurodevelopmental disorder...” in the first sentence of the Abstract and Introduction is a bit redundant. Consider changing to “Dyslexia is a neurodevelopmental disorder”.
Abstract:
· Recommend a few modifications to the initial description of dyslexia. Currently: “...a deficit in accuracy and fluency while reading or spelling whilst the individual does not present deficits in other cognitive domains.” Recommend: “... a deficit in accuracy and/or fluency while reading or spelling that is not expected given their level of cognitive functioning.” (This modification would clarify that dyslexia may co-occur with other diagnoses.)
Introduction:
· When describing how dyslexia is defined in the DSM-5 [Pg 1, Ln 23 and Ln 32], it would be more accurate to say that it falls under the umbrella of the diagnosis “specific learning disorder of reading”. The DSM includes specifiers for issues in word reading accuracy, reading rate, and fluency and reading comprehension. Since dyslexia is not defined by a deficit in reading comprehension, it is helpful to clarify the distinction between the two.
· Pg 2, Ln 40. I would be hesitant to present research in clinical neuroimaging as searching for a biomarker replacement to current clinical evaluation protocols. The individual variability in imaging makes prospects of diagnosis from imaging unreliable. It would be more accurate to describe how neuroimaging is being used to understand the biological mechanisms behind reading development and dyslexia.
· The description of DWI/DTI is a bit difficult to follow. To aid readers unfamiliar with DTI, it may be helpful to first describe that it is a measure of directionality of water droplets and specify the different metrics obtained. Then, those directionality metrics are used as a proxy for tractography and myelination.
· In discussing use of DTI in diagnostics, it would be helpful to clarify what type of disorders it is used for (versus those it is not used for).
· Meta-analyses have been done on DTI in reading/dyslexia (e.g., Barquero et al, 22014; Moreau et al, 2018). The findings of these should be discussed followed by a description of how this systematic review adds to that literature.
Methods:
· For exclusion criteria (Pg 3, Ln115), please include information on what minimal outcomes/methods of analysis were required for use.
Results:
· Were any articles from the search excluded due to lack of full text available?
· Pg 5, Ln 175-176 (and all other instances). Assuming that sex defined at birth is being referenced, instead of “both genders”, please use “males and females”. Replace instances of “girl/boy” with “female/male”.
· Pg 10, Ln 112. Delete double “tests” at the end of the sentence.
· Pg 11, Ln 249. Correct typo “Reaching” to “Reading”.
· Pg 17, Ln 262. Correct typo “The most...” to “Most...”.
· Pg 17, Ln 292-299. This run-on sentence needs to be broken down into more concise components.
Discussion:
· Pg 21, Ln 458-462. Punctuation is missing (run-on sentence).
· Pg 23, Ln 564. The limitations paragraph reads more like a list of issues and does not inform the reader as to why the limitations are important and how to correct them in the future. For example, discussion of the first limitation could be enhanced by conceptualizing it as differences in how dyslexia was defined across studies & acknowledging how this might have impacted the findings.
Conclusions:
· Pg 23, Ln 572. Please expand from this single, run-on sentence. E.g., why is it important to know these things and how will it impact future work?
Tables/Figures:
· Table 1. Appreciate all of the detail that was fit into this table! Please replace “Gender (W:M)” with “Sex (F:M)”.
Citations Referenced:
Barquero LA, Davis N, Cutting LE. Neuroimaging of reading intervention: a systematic review and activation likelihood estimate meta-analysis. PLoS One. 2014 Jan 10;9(1):e83668. doi: 10.1371/journal.pone.0083668. PMID: 24427278; PMCID: PMC3888398.
Moreau, D., Stonyer, J. E., McKay, N. S., & Waldie, K. E. (2018, Mar 15). No evidence for systematic white matter correlates of dyslexia: An Activation Likelihood Estimation meta-analysis. Brain research, 1683, 36-47. https://doi.org/10.1016/j.brainres.2018.01.014
Author Response
Reviewer #1
This structure literature review focuses on studies using structural neuroimaging (specifically measures of white matter integrity) to compare/contrast participants with and without dyslexia. The methodology and presentation of results are appropriate. However, the manuscript would benefit from enhancements to the Introduction, clarification of some methods and results, and phrasing/grammatical corrections throughout. Please see below for specific comments.
Misc:
- Developmental dyslexia is a neurodevelopmental disorder...” in the first sentence of the Abstract and Introduction is a bit redundant. Consider changing to “Dyslexia is a neurodevelopmental disorder”.
Answer: Thank you for your comments. We changed it. (Pg 1, Ln 1 and 18)
Abstract:
- Recommend a few modifications to the initial description of dyslexia. Currently: “...a deficit in accuracy and fluency while reading or spelling whilst the individual does not present deficits in other cognitive domains.” Recommend: “... a deficit in accuracy and/orfluency while reading or spelling that is not expected given their level of cognitive functioning.” (This modification would clarify that dyslexia may co-occur with other diagnoses.)
Answer: Thank you for your comments. We changed it. (Pg 1, Ln 2)
Introduction:
- When describing how dyslexia is defined in the DSM-5 [Pg 1, Ln 23 and Ln 32], it would be more accurate to say that it falls under the umbrellaof the diagnosis “specific learning disorder of reading”. The DSM includes specifiers for issues in word reading accuracy, reading rate, and fluency and reading comprehension. Since dyslexia is not defined by a deficit in reading comprehension, it is helpful to clarify the distinction between the two.
Answer: Thank you for your comments. We rephrased it. (Pg 1 Ln 23-24)
- Pg 2, Ln 40. I would be hesitant to present research in clinical neuroimaging as searching for a biomarker replacement to current clinical evaluation protocols. The individual variability in imaging makes prospects of diagnosis from imaging unreliable. It would be more accurate to describe how neuroimaging is being used to understand the biological mechanisms behind reading development and dyslexia.
Answer: Thank you for your comments. We modified it as suggested. (Pg 2, Ln 40-42)
- The description of DWI/DTI is a bit difficult to follow. To aid readers unfamiliar with DTI, it may be helpful to first describe that it is a measure of directionality of water droplets and specify the different metrics obtained. Then, those directionality metrics are used as a proxy for tractography and myelination.
Answer: Thank you for your comments. We rewrote it so it could be more clear. (Pg 2, Ln 50-56)
- In discussing use of DTI in diagnostics, it would be helpful to clarify what type of disorders it is used for (versus those it is not used for).
Answer: Thank you for your comments. We added it. (Pg 2, Ln 47-49)
- Meta-analyses have been done on DTI in reading/dyslexia (e.g., Barquero et al, 2014; Moreau et al, 2018). The findings of these should be discussed followed by a description of how this systematic review adds to that literature.
Answer: Thank you for your recommendations. Moreau's paper was included in the text. Barquero's paper is very interesting but is out of our main subject since it is about functional MRI, isn't about dyslexia and our paper does not discuss interventions. (Pg 2, Ln 77-79)
Methods:
- For exclusion criteria (Pg 3, Ln115), please include information on what minimal outcomes/methods of analysis were required for use.
Answer: Thank you for your suggestion. We added it. (Pg 3, Ln 130-133)
Results:
- Were any articles from the search excluded due to lack of full text available?
Answer: No. All articles were found using institutional access or by asking directly to the authors.
- Pg 5, Ln 175-176 (and all other instances). Assuming that sex defined at birth is being referenced, instead of “both genders”, please use “males and females”. Replace instances of “girl/boy” with “female/male”.
- Pg 10, Ln 212. Delete double “tests” at the end of the sentence.
- Pg 11, Ln 249. Correct typo “Reaching” to “Reading”.
- Pg 17, Ln 262. Correct typo “The most...” to “Most...”.
- Pg 17, Ln 292-299. This run-on sentence needs to be broken down into more concise components.
Answer: Thank you for your comments. We corrected all items appointed above. (Pg 4-17, Ln 192-193; 202-203; 218-220; 229-230; 266; 279; 309-316)
Discussion:
- Pg 21, Ln 458-462. Punctuation is missing (run-on sentence).
- Pg 23, Ln 564. The limitations paragraph reads more like a list of issues and does not inform the reader as to why the limitations are important and how to correct them in the future. For example, discussion of the first limitation could be enhanced by conceptualizing it as differences in how dyslexia was defined across studies & acknowledging how this might have impacted the findings.
Answer: Thank you for your observation and suggestion. We corrected all the items highlighted. (Pg 23, Ln 580-595)
Conclusions:
- Pg 23, Ln 572. Please expand from this single, run-on sentence. E.g., why is it important to know these things and how will it impact future work?
Answer: Thank you for your suggestion. We improved the conclusion of the manuscript and added the relevance of review findings to future studies and interventions/treatments for dyslexia in different ages. (Pg 23, Ln 597-604)
Tables/Figures:
- Table 1. Appreciate all of the detail that was fit into this table! Please replace “Gender (W:M)” with “Sex (F:M)”.
Answer: Thank you for your comments. We changed it.
Citations Referenced:
Barquero LA, Davis N, Cutting LE. Neuroimaging of reading intervention: a systematic review and activation likelihood estimate meta-analysis. PLoS One. 2014 Jan 10;9(1):e83668. doi: 10.1371/journal.pone.0083668. PMID: 24427278; PMCID: PMC3888398.
Moreau, D., Stonyer, J. E., McKay, N. S., & Waldie, K. E. (2018, Mar 15). No evidence for systematic white matter correlates of dyslexia: An Activation Likelihood Estimation meta-analysis. Brain research, 1683, 36-47. https://doi.org/10.1016/j.brainres.2018.01.014

Reviewer 2 Report
Comments and Suggestions for Authors
This article is a systematic review. The review seeks to find a better understanding of the many possible methods for analyzing DTI data in a clinical population such as developmental dyslexia.
Since exactly such systematic reviews have not yet been done, I believe that this topic is important to develop and the article addresses a specific gap in the field.
The material in the article shows for what purposes and with what software DTI analysis was performed in the selected studies. This is valuable information that will help researchers who are interested in this topic.
The merit of the article is that the literature strategy is clearly and thoroughly described. The authors provide keywords and bases for searching, as well as criteria for inclusion and exclusion of material for further analysis. PRISMA flowchart is also very good.
The conclusions are consistent with the evidence and arguments presented and they address the main question posed. But they do not address one aspect that was highlighted as a part of the study's purpose: 'a better understanding of the multitude of possible methods'. Please add.
All the references are appropriate.
A few more comments:
In abstract the authors use the abbreviation DTI, but they have not introduced it before. This should be corrected.
The tables in the article are very large and difficult to read. I would recommend making fewer columns.
Author Response
March 17, 2024
brainsci-2918643
Dear Ms. Mina Zhang
Section Managing Editor
We are sending the revised version of the manuscript entitled, “Investigating Dyslexia Through Diffusion Tensor Imaging Across Ages: A Systematic Review” manuscript ID - brainsci-2918643, with point-by-point corrections (see below) suggested by reviewer 2. The changes in the manuscript have been highlighted in blue font.
Thank you again for your time and consideration. We hope the paper is now suitable for publication in Brain Sciences. We look forward to hearing your decision.
Sincerely,
Mariana Penteado Nucci
Reviewer #2
This article is a systematic review. The review seeks to find a better understanding of the many possible methods for analyzing DTI data in a clinical population such as developmental dyslexia.
Since exactly such systematic reviews have not yet been done, I believe that this topic is important to develop and the article addresses a specific gap in the field.
The material in the article shows for what purposes and with what software DTI analysis was performed in the selected studies. This is valuable information that will help researchers who are interested in this topic.
The merit of the article is that the literature strategy is clearly and thoroughly described. The authors provide keywords and bases for searching, as well as criteria for inclusion and exclusion of material for further analysis. PRISMA flowchart is also very good.
Answer: Thank you for your comments
The conclusions are consistent with the evidence and arguments presented and they address the main question posed. But they do not address one aspect that was highlighted as a part of the study's purpose: 'a better understanding of the multitude of possible methods'. Please add.
Answer: Thank you for your comments. We added it, improving this aspect of the conclusion in the manuscript.
All the references are appropriate.
Answer: Thank you for your comments.
A few more comments:
In abstract the authors use the abbreviation DTI, but they have not introduced it before. This should be corrected.
Answer: Thank you for your comments. We corrected it.
The tables in the article are very large and difficult to read. I would recommend making fewer columns.
Answer: Thank you for your recommendation. The tables were planned so they would contain key information to run a DTI analysis either with only diffusion data (Table 2) or with clinically relevant data (Table 1) for correlation in dyslexia. Deleting any columns would deprive readers of important aspects for understanding or planning their own research about it. It is also very common in systematic reviews to contain large tables based on the type of detail this kind of publication requires. Below we attach some systematic reviews from BrainSciences that published similar tables.
https://www.mdpi.com/2076-3425/14/3/248
https://www.mdpi.com/2076-3425/14/3/210
m/2076-3425/14/3/210
